# Choline metabolism underpins macrophage IL-4 polarization and RELMα up-regulation in helminth infection

Peyman Ghorbani[1,2,3☯], Sang Yong Kim[4☯], Tyler K. T. Smith[1,2,3], Lucía Minarrieta[1,2,3], Victoria Robert-Gostlin[1,2,3], Marisa K. Kilgour[1,2], Maja Ilijevska[1,2], Irina Alecu[1,2,3], Shayne A. Snider[1], Kaitlyn D. Margison[1], Julia R. C. Nunes[1,2,3], Daniel Woo[4], Ciara Pember[1,2], Conor O'Dwyer[1,2,3], Julie Ouellette[5,6], Pavel Kotchetkov[5,6], Julie St-Pierre[1,2,3], Steffany A. L. Bennett[1,2,3,7,8], Baptiste Lacoste[5,6,8], Alexandre Blais[1,2,3,8,9], Meera G. Nair[4]*, Morgan D. Fullerton[1,2,3,7]*

1 Department of Biochemistry, Microbiology and Immunology, Faculty of Medicine, University of Ottawa, Ottawa, Ontario, Canada, 2 Centre for Infection, Immunity and Inflammation, University of Ottawa, Ottawa, Ontario, Canada, 3 Ottawa Institute of Systems Biology, University of Ottawa, Ottawa, Ontario, Canada, 4 Division of Biomedical Sciences, School of Medicine, University of California Riverside, Riverside, California, United States of America, 5 Neuroscience Program, Ottawa Hospital Research Institute, Ottawa, Ontario, Canada, 6 Department of Cellular and Molecular Medicine, Faculty of Medicine, University of Ottawa, Ottawa, Ontario, Canada, 7 Centre for Catalysis Research and Innovation, University of Ottawa, Ottawa, Ontario, Canada, 8 University of Ottawa Brain and Mind Institute, University of Ottawa, Ottawa, Ontario, Canada, 9 Éric Poulin Centre for Neuromuscular Disease, Ottawa, Ontario, Canada

☯ These authors contributed equally to this work.
* meera.nair@medsch.ucr.edu (MGN); morgan.fullerton@uottawa.ca (MDF)

## Abstract

Type 2 cytokines like IL-4 are hallmarks of helminth infection and activate macrophages to limit immunopathology and mediate helminth clearance. In addition to cytokines, nutrients and metabolites critically influence macrophage polarization. Choline is an essential nutrient known to support normal macrophage responses to lipopolysaccharide; however, its function in macrophages polarized by type 2 cytokines is unknown. Using murine IL-4-polarized macrophages, targeted lipidomics revealed significantly elevated levels of phosphatidylcholine, with select changes to other choline-containing lipid species. These changes were supported by the coordinated up-regulation of choline transport compared to naïve macrophages. Pharmacological inhibition of choline metabolism significantly suppressed several mitochondrial transcripts and dramatically inhibited select IL-4-responsive transcripts, most notably, *Retnla*. We further confirmed that blocking choline metabolism diminished IL-4-induced RELMα (encoded by *Retnla*) protein content and secretion and caused a dramatic reprogramming toward glycolytic metabolism. To better understand the physiological implications of these observations, naïve or mice infected with the intestinal helminth *Heligmosomoides polygyrus* were treated with the choline kinase α inhibitor, RSM-932A, to limit choline metabolism *in vivo*. Pharmacological inhibition of choline metabolism lowered RELMα expression across cell-types and tissues and led to the disappearance of peritoneal macrophages and B-1 lymphocytes and an influx of infiltrating monocytes. The impaired macrophage activation was associated with some loss in optimal immunity to *H. polygyrus*, with increased egg burden. Together, these data demonstrate that choline metabolism is

**Data Availability Statement:** All data needed to evaluate the conclusions in the paper are present in the paper and/or the Supplementary Materials. Raw and processed RNA sequences have been

deposited in the NCBI GEO database under accession GSE236252.

**Funding:** This research was supported by a Discovery Grant from the Natural Sciences and Engineering Research Council of Canada (NSERC, RGPIN-2021-03503 to M.D.F.), the Canadian Institutes of Health Research (CIHR; PJT-179853 to S.A.L.B., PJT-183839 to A.B. and PJT-148634 to M.D.F.) and National Institutes of Health (NIAID, R01AI153195 to M.G.N.). M.D.F. holds a Camille Villeneuve Chair in Cardiovascular Immunometabolism and was supported by a CIHR New Investigator Award (MSH-141981). J.S.P. holds a Canada Research Chair in Cancer Metabolism (Tier 1). T.K.T.S was supported by a CIHR Vanier Scholarship, S.A.S. was supported by an NSERC Canadian Graduate Scholarship, J.R.C.N and C.P. were supported by Ontario Graduate Scholarships and J.O. is supported by a CIHR Doctoral Research Award. P.K. is supported by a doctoral training award from uOttawa Brain and Mind Research Institute. The funders had no role in study design, data collection and analysis, decision to publish, or preparation of the manuscript.

**Competing interests:** The authors have declared that no conflict of interest exists.

required for macrophage RELMα induction, metabolic programming, and peritoneal immune homeostasis, which could have important implications in the context of other models of infection or cancer immunity.

## Author summary

Metabolic factors such as diet can profoundly impact the immune response to pathogens. Choline is an essential nutrient that functions as a precursor to membrane phospholipids. Here, we identify a critical role for choline metabolism in driving IL-4 macrophage polarization and shaping the cellular immune response to intestinal helminth infection. IL-4 polarized macrophages had increased phosphatidylcholine biosynthesis and content. Inhibition of choline metabolism dramatically impaired macrophage activation and metabolism in culture, and *in vivo* following intestinal helminth infection, which was accompanied by reduced expression of secreted protein RELMα, and a change in the immune cellular composition. Overall, these studies identify a previously unrecognized role for the essential nutrient choline in shaping the cellular immune landscape under normal conditions and in response to intestinal helminths.

## Introduction

Macrophages represent a heterogeneous and plastic subset of innate immune cells that perform critical homeostatic, surveillance and effector tasks in almost all tissues [1]. They can dynamically respond to diverse endogenous and exogenous cues, including nutrient and energy availability. Metabolic programming can directly drive macrophage function but can also be rewired in response to various stimuli [2]. While we continue to expand our understanding of how energy-generating pathways and their intermediates underpin immunometabolic function, the importance of nutrients in macrophage biology remains largely underexplored.

Choline is a quaternary amine and essential nutrient that can form the main eukaryotic phospholipid class, glycerophosphocholines (the most abundant of which are phosphatidylcholines (PC)), or form acetylcholine in cholinergic tissues [3]. Additionally, methyl groups from choline contribute to one-carbon metabolism after its mitochondrial oxidation. Given its positive charge, choline requires facilitated transport across cellular membranes, representing the first but least studied step in its metabolism. In immune cells, choline transporter-like protein 1 (Slc44a1) is responsible for the majority of choline uptake [4, 5]. Upon uptake, choline is thought to be rapidly phosphorylated by choline kinase (Chkα/Chkβ) and is then committed to the PC biosynthetic (CDP-choline/Kennedy) pathway [3, 6].

Phospholipid biosynthesis is indispensable for membrane biogenesis, which plays an important role in cellular differentiation and phagocytosis [7]. The first insight into the potential role of choline in macrophage function came over twenty years ago with the observation that deletion of myeloid *Pcyt1a*, encoding the rate-limiting enzyme in PC synthesis, diminished adaptive responses to endogenous stimuli (free cholesterol) [8] and impaired the secretion of pro-inflammatory cytokines from macrophages [9]. Recent lipidomic analyses of cultured macrophages demonstrated sweeping alterations to phospholipid composition and content in response to various inflammatory stimuli [4, 10–12]. To begin to address the specific roles of choline as a regulator of macrophage functions, we and others have shown that

polarization with the bacterial endotoxin lipopolysaccharide (LPS; termed M[LPS]) in primary mouse macrophages leads to increased choline uptake and PC synthesis in an *Slc44a1*-dependent manner. Inhibiting choline uptake and metabolism led to more pronounced inflammatory responses in M[LPS], but also diminished inflammasome-mediated cytokine release [4, 5]. How choline metabolism may be regulated by or be important for macrophage phenotypes in response to Th2 cytokine exposure remains unknown.

Interleukin-4 (IL-4), along with IL-13, is a signature cytokine of type 2 inflammatory responses [13]. In addition to being widely used to polarize macrophages *in vitro* (termed M [IL-4]), secretion of these cytokines *in vivo* is a critical response triggered by invading macroparasites or allergens [14]. Since it is now appreciated that macrophage inflammatory and metabolic programs exhibit a dichotomy in response to prototypical pro- and anti-inflammatory stimuli, we sought to characterize the interplay between choline metabolism and IL-4-mediated polarization of macrophages. Interestingly, we found that identical to LPS, choline uptake and incorporation into PC was augmented in M[IL-4]. Moreover, when choline uptake and metabolism was inhibited, we observed select but striking differences in metabolic and IL-4-specific responses, underpinned by a dramatic reduction in RELMα both *in vitro* and in mouse parasitic helminth infection.

## Results

### IL-4 increases PC content and alters choline containing phospholipid composition

To begin to interrogate the potential effect of IL-4 stimulation on choline metabolic pathways, we performed targeted LC-MS/MS analysis of choline-containing phospholipids in naïve (M [0]), M[IL-4] and M[LPS] (Fig 1A). Surprisingly, the typically divergent polarization states led to a symmetrical increase in total cellular PC levels (Fig 1B) with a significant increase predominantly in PCs containing unsaturated and monounsaturated fatty acids (Fig 1C). With respect to other choline-containing lipids, no differences were found in the level of sphingomyelin (Fig 1D). Interestingly, there were also changes within the choline-containing lipidome in response to either IL-4 or LPS. The plasmalogen forms of PC, which have been demonstrated to be important for downstream lipid signaling, showed divergent changes. Levels of plasmenyl PCs (PC(P)), structurally known as 1-alkenyl,2-acylglycerophosphocholines, as well as their immediate precursors (LPC(P)) were reduced in response to both stimuli, while levels of plasmanyl PCs (PC(O)), known as 1-alkyl,2-acylglycerophosphocholines remained unchanged (Figs 1E, 1F, and S1A). The levels of platelet activating factors (PAF), their immediate precursor the lyso-PAFs (LPC(O)) (S1B and S1C Fig), as well as total lysophosphatidylcholines (LPCs) (Fig 1G) did not change in response to IL-4 but were significantly up-regulated in M[LPS]. The fatty acid composition of choline-containing subspecies was indistinguishable between LPS and IL-4 polarization, except for a divergence in total LPCs (S1D Fig).

### IL-4 up-regulates choline metabolism in macrophages

In keeping with higher levels of total cellular PC, polarization with IL-4 increased the incorporation of choline into PC (Fig 1H), as well as the rate of choline uptake in bone marrow-derived macrophages (BMDM) (Fig 1I), without changing choline transport affinity or sensitivity to inhibition (S2A and S2B Fig). Moreover, while select phospholipases are up-regulated in M[IL-4] [15] we observed no difference in the rate of PC degradation (S2C Fig), which completely mirrored the effects of LPS polarization [4]. We and others have shown that macrophages express two choline transporters, Slc44a1 and Slc44a2. IL-4-treated cells did not show

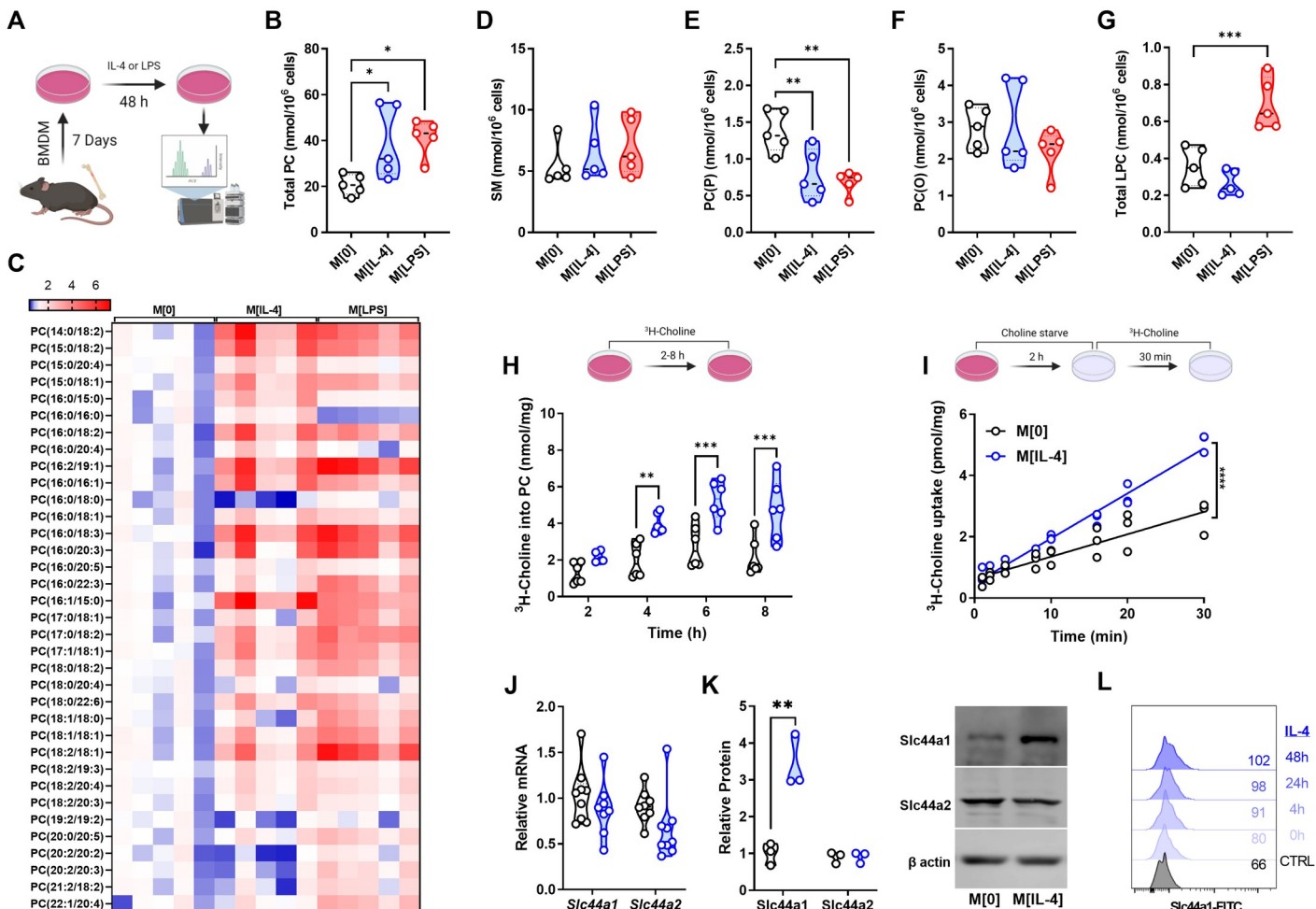

**Fig 1. IL-4 up-regulates choline metabolism in macrophages.** A) Schematic of lipidomics analysis. B-G) Total PC content (B), heatmap of PC species (C), SM content (D), PC(O) content (E), PC(P) content (F), or total LPC content (G) for M[0], M[IL-4], or M[LPS] expressed as nmol per 10⁶ cells. n = 5 per polarization. Heatmap statistics are shown as fold change over the average of M[0]. One-way ANOVA with Dunnett's test for multiple comparisons (* p < 0.05, ** < 0.01). H) Incorporation of ³H-choline into phospholipids over time. n = 5 per timepoint, representative of 3 experiments. Two-way ANOVA with Šídák's test for multiple comparisons (** p < 0.01, *** p < 0.001). I) Uptake of ³H-choline over time in M[0] or M[IL-4]. n = 3 per timepoint, representative of 3 experiments. Linear regression F test (**** p < 0.0001). J) Relative expression of *Slc44a1* and *Slc44a2* transcripts in M[0] or M[IL-4], normalized to *Actb*. n = 8 mice. K) Expression by Western blot of Slc44a1 and Slc44a2 in M[0] or M[IL-4]. Densitometry quantified relative to β-actin. n = 3, representative of 3 experiments. Unpaired t test (** p < 0.01). L) Expression of Slc44a1 by surface flow cytometry after different IL-4 stimulation times. Representative of 2 experiments. Schematics were created using BioRender.

increased *Slc44a1* or *Slc44a2* mRNA (Fig 1J); however, total (Fig 1K) and surface (Fig 1L) Slc44a1 protein expression was significantly augmented, suggesting post-transcriptional mechanisms. There were no changes in Slc44a2 protein levels. Taken together, these data suggest a coordinated up-regulation of choline uptake and subsequent incorporation into PC in response to IL-4 that is comparable in magnitude to LPS stimulation, though subtly different in terms of mechanism.

## Inhibiting choline metabolism selectively impairs IL-4 polarization

Characteristically, IL-4 polarization induces the expression of genes involved in dampening inflammation and promoting tissue repair. We next sought to understand how limiting

choline availability and metabolism may affect these well-known responses. We used an inhibitor of extracellular choline uptake (hemicholinium-3; HC3) [16] and an inhibitor of choline incorporation into PC, the choline kinase inhibitor RSM-932a (RSM) [17]. BMDM incubated with HC3 or RSM for 24 h prior to polarization had little effect on most hallmark M[IL-4] genes, such as *Arg1*, *Mrc1* and *Chil3* (Fig 2A–C); however, completely suppressed IL-4-mediated induction of *Retnla* (Figs 2D and S3A–S3C). Interestingly, shorter (6 h) treatment with HC3 was insufficient to affect *Retnla* expression (S3D Fig). To ascertain a better understanding of the kinetics of *Retnla* inhibition, we conducted a simple experiment in which cells were treated with inhibitors for 24 h, followed by IL-4 for a further 24 h (labeled POST), 24 h of concurrent inhibitor and IL-4 treatment (labeled as CONC) and 24 h of IL-4 polarization, followed by 24 h of inhibition (labeled as PRE). Inhibiting choline metabolism prior to IL-4 polarization flattened any induction of *Retnla*. Concurrent inhibition of choline uptake and metabolism along side polarization also significantly limited *Retnla* up-regulation. Finally, while inhibition of choline uptake with HC3 in already polarized macrophages tended to diminish expression, blocking choline metabolism with RSM in polarized cells was sufficient to completely suppress *Retnla* expression (Fig 2E). To begin to explain the select effect of choline inhibition in M [IL-4], we assessed the activation of pSTAT6-Y641 after IL-4 stimulation, which was significantly blunted by treatment with HC3 or RSM after 15 minutes (Fig 2F). There were no changes in levels of *Il4ra* transcript (S3E Fig), and pSTAT6-Y641 phosphorylation was further diminished over time (S3F Fig). Together, this suggests that disruptions of choline metabolism in macrophages can blunt STAT6 signaling, though it is likely that ligand recognition through IL-4R is not impaired [18].

To further probe how limiting choline metabolism alters macrophage IL-4 polarization, we used CD38 and Egr2, which have been identified as effective discriminators of M [LPS] vs. M [IL-4], respectively [19]. Blocking choline uptake did not impair the induction of Egr2, nor did it promote CD38 expression in M [IL-4] (S3G Fig), despite Egr2 being induced in an IL-4/STAT6-dependent manner [20]. We next assessed the expression of the checkpoint ligands PD-L1 and PD-L2, which have also been used discriminate between M [LPS] and M [IL-4] polarization [21]. Inhibiting choline metabolism with either HC3 or RSM led to consistent PD-L1 up-regulation in M[0] and M[IL-4], but not in M[LPS] where expression was already maximal (Fig 2G). In contrast, inhibiting choline metabolism reduced PD-L2 expression in M [IL-4] (Fig 2G). Therefore, by limiting uptake and phosphorylation of choline, there were pointed changes to traditional IL-4-induced cytokines (e.g., *Retnla*) and a skewing toward a M [LPS]-like profile.

## Choline metabolism inhibition impairs macrophage function

The secreted cytokine resistin-like molecule alpha (RELMα) is encoded by *Retnla* and has pleiotropic functions, including suppression of Th2 responses and promotion of wound healing [22]. In IL-4-polarized macrophages, blocking choline uptake significantly reduced intracellular RELMα levels (Fig 2H). Furthermore, to rule out off-target pharmacological effects, culture of M [IL-4] in choline-deficient media also curtailed intracellular RELMα to the same extent as HC3 and RSM (Fig 2H). Finally, we confirmed that secreted RELMα in M[IL-4] conditioned media was significantly lower when choline uptake was inhibited (Fig 2I). Functionally, given that RELMα levels were dramatically reduced in response to altered choline uptake and metabolism, we sought to assess the *in vitro* wound healing capacity of macrophages. A "scratch wound" was created among 3T3-L1 fibroblasts grown to confluency and when exposed to conditioned media from HC-3-treated M[IL-4], we observed a significantly impaired rate of wound healing (S3H Fig).

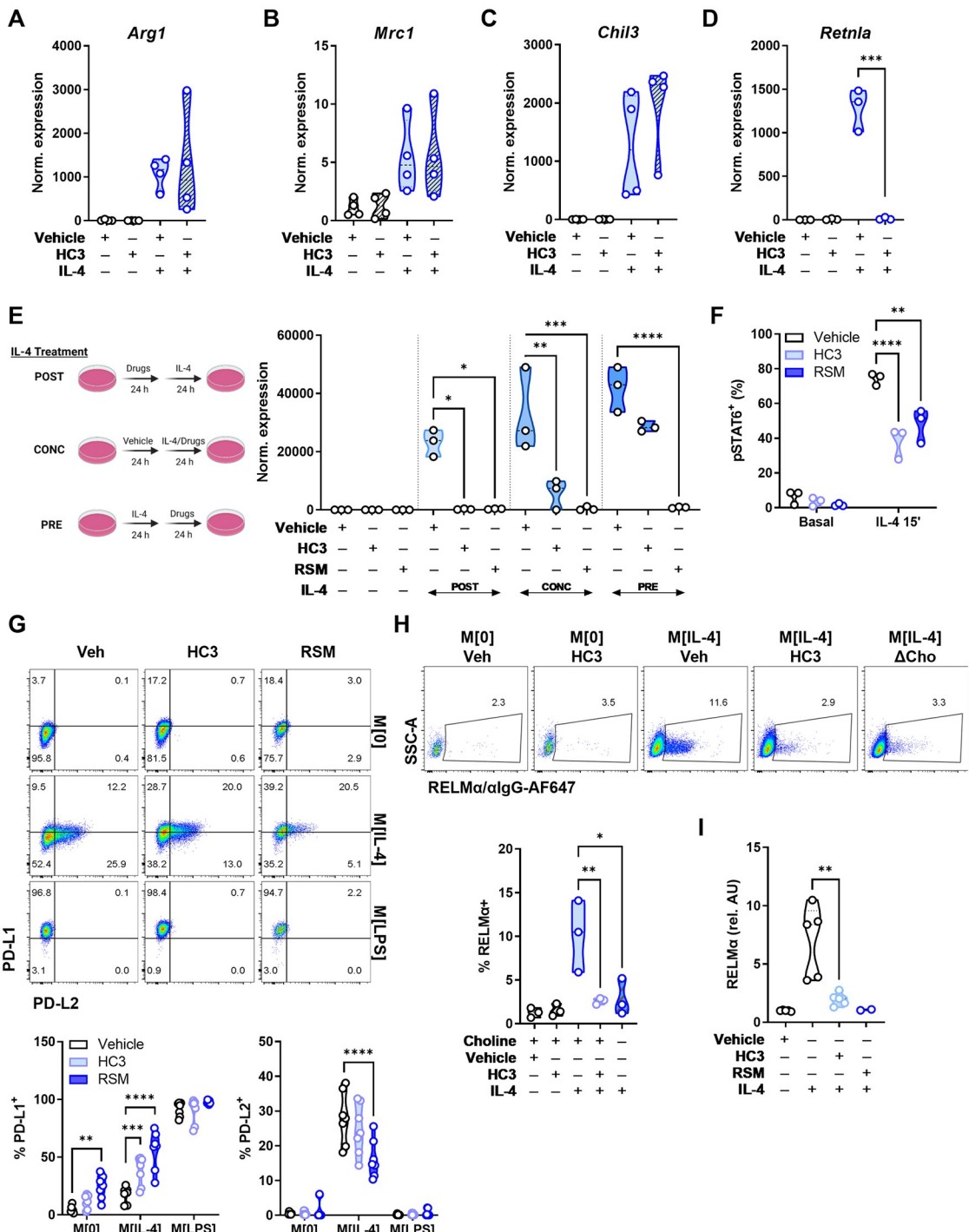

**Fig 2. Choline uptake and phosphorylation is required for normal IL-4 signaling and M[IL-4] phenotype.** A-D) Macrophages were treated with vehicle (DMSO) or HC3 (250 μM) for 24 h, washed, then treated with IL-4 (20 ng/mL) for 24 h. Relative expression of M[IL-4] hallmark genes *Arg1*, *Mrc1*, *Chil3*, or *Retnla* normalized to *Actb* and compared to M[0]. n = 3–4, representative of 3–5 experiments. Unpaired t test (*** p < 0.001). E) Schematic of timing variations for inhibitors and IL-4 polarization. Macrophages were treated with vehicle (DMSO), HC3 (250 μM), or RSM (5 μM) in the first 24 h then polarized with IL-4 (20 ng/mL) for 24 h (POST). Inhibitors were given together with IL-4 for 24 h (CONC). Macrophages were polarized with IL-4 for 24 h, then treated with inhibitors for 24 h (PRE). Expression of *Retnla* normalized to *Actb* and compared to M[0]. n = 3, representative of 1 experiment. Two-way ANOVA with Tukey's test for multiple comparisons (* p < 0.05, ** p < 0.01, *** p < 0.001, **** p < 0.0001). F) Macrophages were treated with vehicle (DMSO), HC3 (250 μM) or RSM-932a (5 μM) for 24 h, washed, then treated with IL-4 (20 ng/mL) for 15' and stained for intracellular pSTAT6 (Tyr641). n = 3, representative of 3 experiments. Two-way

ANOVA with Tukey's test for multiple comparisons (** p < 0.01, **** p < 0.0001). G) Surface PD-L1 and PD-L2 expression in M [0], M[IL-4], or M[LPS], treated with vehicle, HC3 (250 μM) or RSM-932a (5 μM). n = 2, representative of 3 experiments. H) Macrophages were treated with vehicle (DMSO) or HC3 (250 μM) for 24 h, washed, then treated with IL-4 (20 ng/mL) for 24 h in complete or choline-deficient (ΔCho) media. B) Quantification of intracellular RELMα staining in live F480$^+$ macrophages. n = 3, representative of >4 experiments. One-way ANOVA with Tukey's test for multiple comparisons (* p < 0.05, ** p < 0.01). I) Macrophages were treated with vehicle (DMSO), HC3 (250 μM), or RSM-932a (5 μM) for 24 h, washed, then treated with IL-4 (20 ng/mL) for 24 h. Detection of supernatant RELMα by ELISA. n = 5 (vehicle, HC3, IL-4) or n = 2 (RSM). Unpaired t test (** p < 0.01). Schematics were created using BioRender.

## Choline metabolism supports naïve and M[IL-4] transcriptional programs

To gain an unbiased understanding of the transcriptional changes in response to blocking choline uptake and metabolism, we next performed RNA transcriptomic analysis. Naïve and IL-4-polarized macrophages that had been pretreated with either HC3 or RSM prior to polarization showed profound and sweeping differences in transcript expression patterns. Reactome pathway analysis revealed the up-regulation of extracellular matrix organization genes by choline inhibition, and down-regulation of genes involved in GPCR signalling (M[0]) and cell cycle (M[IL-4]) (S4 Fig). Importantly, we confirmed that *Retnla* was only expressed with IL-4 polarization and was one of the most strongly down-regulated gene by fold change compared to vehicle for both HC3 and RSM treatments (Fig 3A). We next performed Enrichr-KG analysis [23] on the top 100 shared genes up- and down-regulated by HC3 and RSM in both M[0] and M[IL-4]. These analyses indicated a conserved induction of unfolded protein response and ER stress pathway genes, such as *Trib3*, *Ern1*, *Ddit3* (Fig 3B and 3C). In contrast, both drugs down-regulated genes involved in tyrosine phosphorylation and DNA replication (Fig 3D).

## Choline metabolism inhibition profoundly alters mitochondrial morphology and function

Macrophage metabolism is highly responsive to exogenous stimuli [24]. In response to IL-4, glycolysis, glutaminolysis, and oxidative phosphorylation are up-regulated [25]. The inhibition of choline metabolism led to a dramatic decrease in select electron transport chain (ETC) genes encoded in the mitochondrial genome, with concomitant up-regulation of two mitochondrial ribosomal genes (*mt-Rnr1*/*mt-Rnr2*) (Fig 4A). Immunoblot analysis of ETC complexes showed a strong trend toward reductions in complexes I-IV with HC3 and RSM (Fig 4B), though only changes in complex I was significantly different (S5A Fig). To interrogate further, we performed extracellular flux analysis and as expected, oxygen consumption rate (OCR) and spare respiratory capacity (SRC) were increased in M[IL-4] compared to M[0] (Fig 4C-I). Treatment with HC3 or RSM resulted in a potent reduction of SRC (Fig 4H). Presumably as a compensatory mechanism, glycolysis as measured by extracellular acidification rate (ECAR) was modestly elevated by HC3 and highly increased by RSM (Fig 4D and 4F), effects reminiscent of complex I inhibition. We used a recently refined method to calculate energy derived from glycolysis ($ATP_{Glyco}$) or oxidative phosphorylation ($ATP_{OXPHOS}$) using extracellular flux assay values [26]. This showed that although $ATP_{OXPHOS}$ was diminished with HC3 or RSM, overall ATP production remained relatively stable (Fig 4J). To understand the kinetics of these effects, we next assessed the time needed for metabolic changes in response to choline kinase inhibition. Prior to washing the assay plate, just 5 minutes of RSM treatment was sufficient to partially reduce $ATP_{OXPHOS}$ production, while 1 hour of treatment strongly reduced $ATP_{OXPHOS}$ (S5B Fig). However, direct injection of RSM instead of rotenone/antimycin A failed to reduce OCR (S5C Fig). This suggests that while rapid, the inhibitory effect on oxidative metabolism is indirect by way of general mitochondrial disruption rather than a direct off-target activity on ETC complexes.

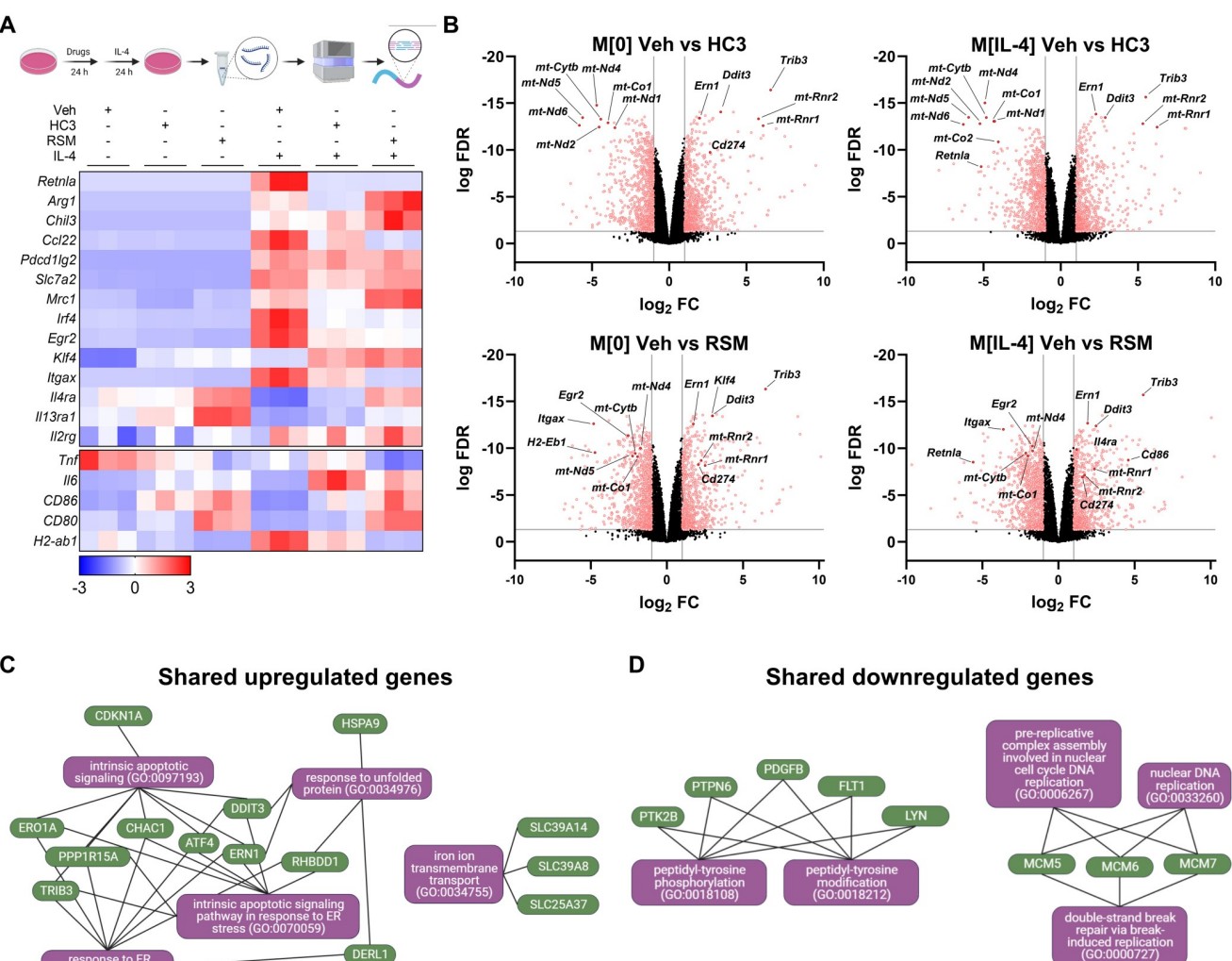

**Fig 3. Inhibiting choline metabolism drives pronounced changes in gene transcription in naïve and IL-4-polarized macrophages.** A) Schematic of bulk RNA-seq sample preparation. Heatmap of z-scores for select genes of interest. n = 3. B) Volcano plots of comparisons between Veh and HC3 or RSM in M [0] or M[IL-4]. Red dots represent genes with false discovery rate (FDR) < 0.05 and $\log_2$ fold change (FC) < -1 or > 1. Select genes of interest are annotated. C-D) The top significantly up- or down-regulated genes shared in all comparisons in B) were analyzed by Enrichr-KG to identify common pathways. Schematics were created using BioRender.

TEM images of RSM-treated M[IL-4] displayed a near complete lack of intact mitochondria with normal cristae structure (Fig 4K). We also observed examples of engulfed mitochondria in RSM-treated M[IL-4] cells, indicative of mitophagy as seen by others [5]. In line with this and previous reports, choline metabolism inhibition also impaired mitochondrial membrane potential (S5D Fig). Together, these results suggest that choline inhibition causes a rapid, though not immediate, deterioration of mitochondrial health that culminates in near complete mitochondrial dysfunction.

## Choline kinase inhibition in vivo reduces tissue-resident macrophage RELMα

RELMα is strongly up-regulated upon IL-4 treatment *in vitro* but is also expressed endogenously in tissue-resident macrophages throughout the body, especially serosal cavity

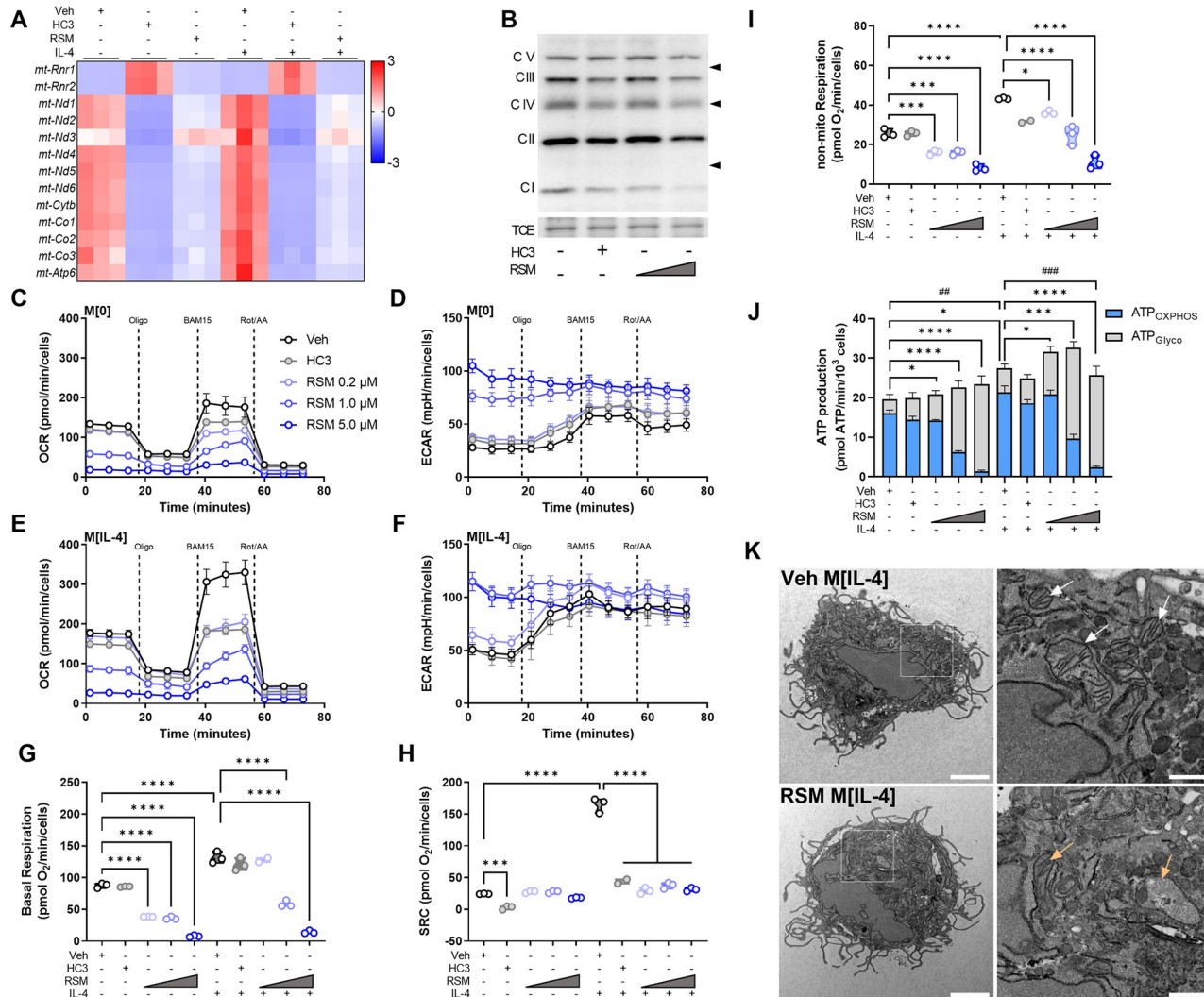

**Fig 4. Inhibiting choline metabolism impairs mitochondrial structure and function.** A) Heatmap of z-scores for select mitochondrially-encoded genes. B) Western Blot of electron transport chain complexes I to V in macrophages treated with vehicle (DMSO), HC3 (250 μM), or RSM (1 or 5 μM). Total protein by trichloroethanol (TCE) staining shown for normalization. Representative blots of n = 3. C-I) Mito Stress Test assay of extracellular flux with sequential treatments of 1.5 μM oligomycin, 14 μM BAM15, and 1 μM rotenone/1 μM antimycin A/Hoechst 33342. Oxygen consumption rate (OCR) of M[0] (C) or M[IL-4] (E) treated with DMSO (Veh), HC3 (250 μM), or different concentrations of RSM. Extracellular acidification rate (ECAR) in M[0] (D) or M[IL-4] (E). Derived parameters of basal respiration (G), spare respiratory capacity (H), or non-mitochondrial respiration (I) from (C) and (E). Measurements (n = 3 in triplicate) were normalized to an arbitrary cell factor determined by Hoechst 33342 nuclei counts. One-way ANOVA with Šidák's test for multiple comparisons (*** p < 0.001, **** p < 0.0001). J) Bioenergetics analysis of ATP produced through oxidative phosphorylation (ATP$_{OXPHOS}$) or glycolysis (ATP$_{Glyco}$). Measurements (n = 3 in triplicate) were normalized per $10^3$ cells. Mixed-effects analysis with Tukey's test for multiple comparisons (ATP$_{OXPHOS}$: ## p < 0.01, ### p < 0.001; ATP$_{Glyco}$: * p < 0.05, *** p < 0.001, **** p < 0.0001). K) Transmission electron micrographs of macrophages treated with vehicle (DMSO) or RSM (5 μM) in M[IL-4]. Scale bars represent 2 μM (left) or 0.5 μM (right insets). White arrows indicate healthy mitochondria with intact cristae, orange arrows indicate mitochondria with aberrant membrane structure or absent cristae. Representative images from n = 3 and at least 10 cells imaged per condition. No cells with intact mitochondria were observed with RSM treatment.

macrophages [27, 28]. We next sought to pharmacologically block choline metabolism *in vivo* to interrogate whether this could mimic changes in IL-4 responses observed in cultured cells. Mice were dosed with RSM (3 mg/kg i.p.) every other day for seven days (Fig 5A). Previously, long-term RSM administration in preclinical tumour models resulted in significant reduction in tumour growth [17], highlighting its low toxicity and therapeutic potential. Surprisingly,

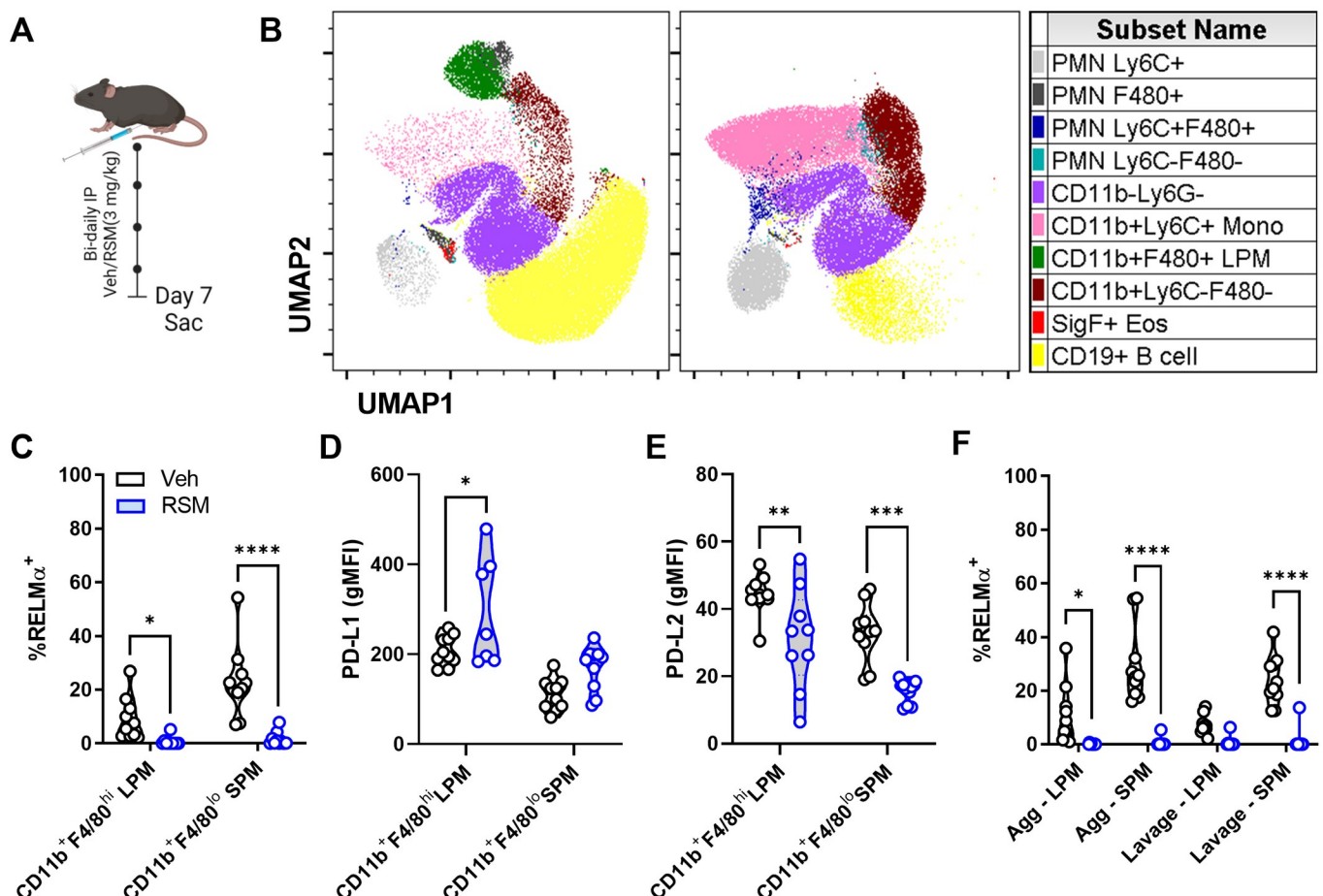

**Fig 5. *In vivo* choline kinase inhibition remodels peritoneal cell populations and impairs RELMα expression.** A) Schematic of 7-day *in vivo* choline kinase inhibition. Mice were treated intraperitoneally with vehicle (40% DMSO in PBS) or RSM-932a (3 mg/kg) every other day for 6 days and sacrificed on day 7. B) UMAP of peritoneal cell populations. n = 9–10, representing 2 independent experiments. C) Intracellular RELMα expression in live CD11b$^+$F4/80$^{hi}$ large (LPM) and CD11b$^+$F4/80$^{lo}$ small (SPM) peritoneal macrophages. Two-way ANOVA with Šídák's test for multiple comparisons (* p < 0.05, **** p < 0.001). D-E) PD-L1 or PD-L2 expression (gMFI) in live CD11b$^+$F4/80$^{hi}$ LPM and CD11b$^+$F4/80$^{lo}$ SPM. Two-way ANOVA with Šídák's test for multiple comparisons (* p < 0.05, ** p < 0.01, *** p < 0.001). F) Intracellular RELMα expression in LPM and SPM isolated from peritoneal cavity aggregates or suspended lavage cells. Two-way ANOVA with Šídák's test for multiple comparisons (* p < 0.05, **** p < 0.0001). Schematics were created using BioRender.

RSM treatment resulted in drastic remodelling of peritoneal exudate cell (PEC) populations (Fig 5B). Uniform manifold approximation and projection plots show a reduction in B cells, macrophages and eosinophils, and an increase in Gr-1$^+$ polymorphonuclear cells (PMNs) and monocytes. Furthermore, intracellular RELMα levels were significantly decreased in large (CD11b$^+$F4/80$^{hi}$), and small (CD11b$^+$F4/80$^{lo}$) PEC macrophages (LPM and SPM) taken from RSM-treated mice compared to vehicle control (Figs 5C and gating in S6A). Moreover, expression of PD-L1 was also significantly increased and PD-L2 reduced compared to vehicle-treated mice, which mirrored *in vitro* results (Fig 5D and 5E). The suppressive effect of RSM on LPM and SPM, which contained moderate or high levels of intracellular RELMα was evident following just two doses, separated by 48 h (S6B and S6C Fig).

Inflammation, infection, or sterile injury of serosal (peritoneal and pleural) cavities cause the disappearance of LPM from serosal fluids (termed macrophage disappearance reaction or macrophage disturbance of homeostasis reaction (MDHR) [29]. Several recent studies have shown that these macrophages are primarily forming aggregates within the cavity or localizing

to the omentum [30]. We adapted a recently published comprehensive methodology [30, 31] to address MDHR in our studies by analyzing potential sites of LPM egress: floating and adherent cellular aggregates, homing to the omentum, and migration to draining lymphatics. We found that aggregated MHCII$^{lo}$ LPM and MHCII$^{+}$ SPM in RSM-treated mice expressed nearly no RELMα, similar to suspended lavage cells (Fig 5F). Thus, *in vivo* inhibition of choline metabolism restructures endogenous tissue-resident macrophage phenotypes and supresses RELMα in multiple tissue compartments.

RELMα is further found in white adipose tissue (WAT) and lung, and though previously thought to be an adipokine [32], RELMα is mainly produced by WAT macrophages. Short-term RSM treatment increased WAT *Emr1* expression (S6D Fig) and matched the pattern of *in vitro* choline uptake or Chkα inhibited BMDM with decreased *Retnla*, increased *Chil3*, and unchanged *Mrc1* (S6E–S6G Fig).

## Inhibiting choline metabolism with RSM improves measures of chemical-induced colitis

The link between choline metabolism and RELMα was unexpected. To further probe the functional consequences of altering RELMα levels under pathophysiological conditions, we used dextran sodium sulfate (DSS) to chemically induce a state of colitis in female C57BL/6J mice over a 7-day protocol, where mice were given vehicle or RSM (3 mg/kg i.p.) bi-daily (S6H Fig). While there were no significant differences in body weight (S6I Fig), colon length was significantly longer and hemoccult was significantly lower in RSM-treated mice compared to control, respectively (S6J and S6K Fig). This is entirely consistent with previous reports where *Retnla*-deficient mice were protected against DSS-induced colitis [33].

## Choline inhibition shifts peritoneal immunity and impairs M[IL-4] polarization to multiple helminth infections

Our data suggest a role for choline metabolism in supporting IL-4-induced (Th2) responses *in vitro* and *in vivo*. To interrogate choline metabolism function in the physiologic setting of a Th2-skewed environment driven by helminth infection, we infected female mice with *Heligmosomoides polygyrus* (*Hp*), a parasitic nematode that colonizes the small intestine and drives protective M[IL-4]-like responses [34]. Intestinal *Hp* infection induces significant peritoneal cavity inflammation and macrophage M[IL-4] responses [35]. The striking RSM-induced changes in the peritoneal immune landscape of naïve mice was exacerbated during *Hp* infection (Figs 6A and S7A). We detected more than a 10-fold increase in peritoneal cells compared to naïve mice and during infection, there was a significantly lower number total cells when mice were treated with RSM (Fig 6B). There was a substantial drop in eosinophils and B-1 cells in RSM-treated mice, accompanied by a rise in monocytes and neutrophils, both by frequencies and total cell numbers (Fig 6C–6F). Peritoneal macrophages were also decreased by RSM treatment in both naïve and infected mice (Fig 6G). Bulk peritoneal macrophages showed differences in CD206, CD86, and PD-L1 in infected mice treated with RSM (Fig 6H–6J), similar to immunofluorescent staining and in naïve mice (Fig 5D). For acute *Hp*-infected mice, we further refined peritoneal macrophage classification into MHCII$^{hi}$ SPM and MHCII$^{lo}$ LPM. The numbers and frequencies of MHCII$^{hi}$ and MHCII$^{lo}$ peritoneal macrophages were decreased upon RSM treatment (S7B Fig). CD86 and PD-L1 were increased whereas CD206 was up-regulated in both populations, with a stronger effect seen in MHCII$^{lo}$ macrophages (S7C–S7E Fig). In contrast, PD-L2 was decreased only in MHCII$^{hi}$ macrophages in RSM-treated mice (S7F Fig). These results were directly recapitulated during chronic *Hp* infection and RSM treatment, where we observed differences in resident peritoneal immune

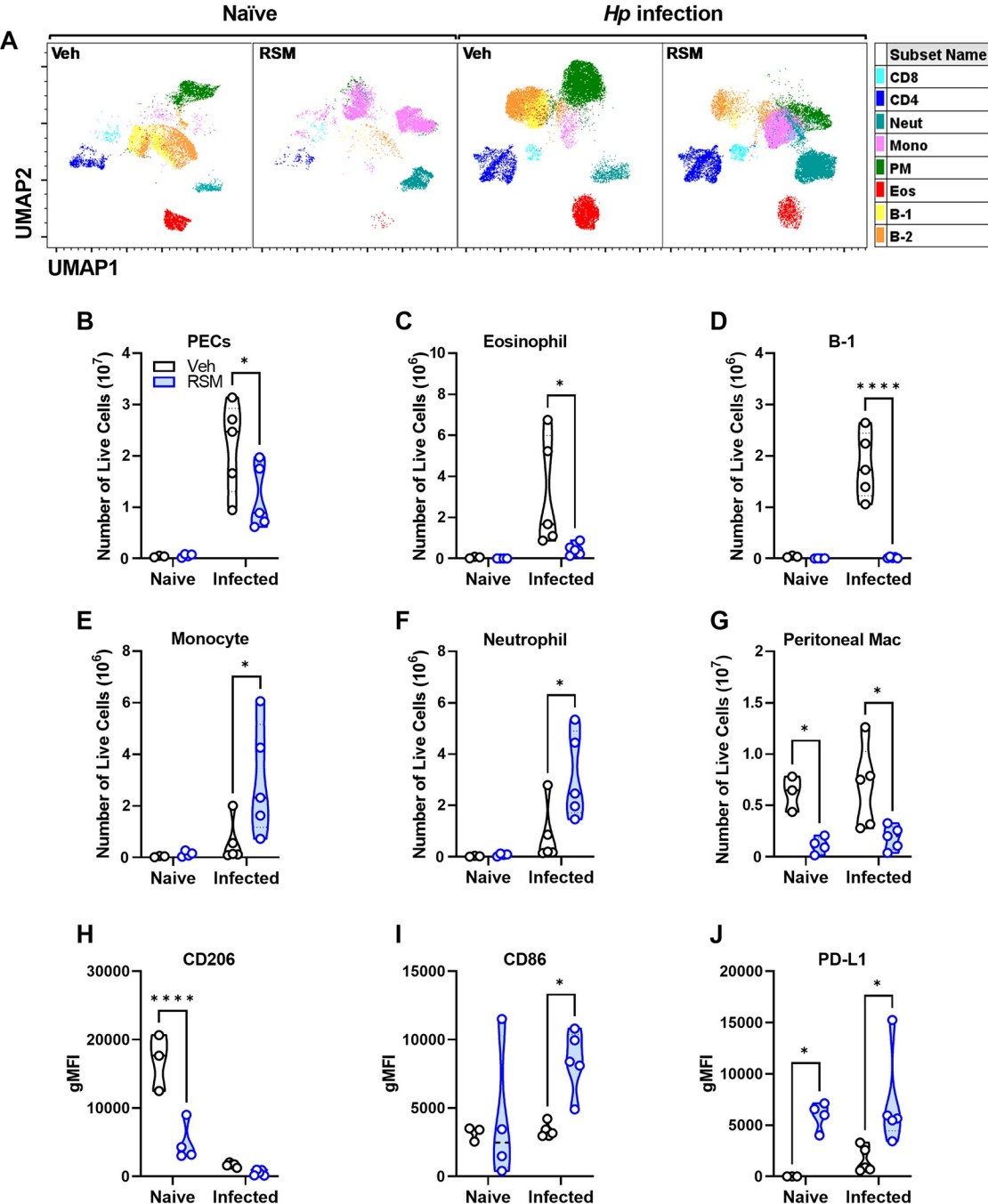

**Fig 6.** *In vivo* **choline kinase inhibition in naïve mice and primary intestinal helminth infection alters peritoneal cell populations and macrophage alternative activation.** A) UMAP plots of peritoneal cells (PECs) from naïve and *H. polygyrus*-infected mice treated with vehicle or RSM-932a. See S6A Fig for population gating. B-G) Enumeration of B) total PECs, C) eosinophils, D) monocytes, E) B-1 cells, F) neutrophils, G) peritoneal macrophages among live PECs. n = 3–4 (naïve) or n = 5 (infected). Two-way ANOVA with Šídák's test for multiple comparisons (* p < 0.05, ** p < 0.01, *** p < 0.001, **** p < 0.0001). H-J) CD206, I) CD86, or J) PD-L1 expression (gMFI) on peritoneal macrophages. n = 3–4 (naïve) or n = 5 (infected). Two-way ANOVA with Šídák's test for multiple comparisons (* p < 0.05), representative of 2 experiments.

populations, as well as changes in macrophage expression of CD86 and CD206 (S8A–S8F Fig). Taken together, these data support a major role for choline metabolism in sustaining RELMα expression and shaping the composition and phenotype of immune populations *in vivo*.

## Choline kinase inhibition does not affect helminth burden despite dramatic blunting of RELMα

Given the dramatic shift in peritoneal immune landscape, we next aimed to assess how markers of parasite burden may be affected. Mice were orally infected with *Hp* larvae and left for 8 days to allow parasites to develop into adults within the intestinal lumen, followed by treatment with RSM or vehicle every two days. Naïve mice were also treated with RSM at the same dose and interval to serve as controls for infection (Fig 7A). Treatment with RSM had modest effects on the weight of naïve or *Hp*-infected mice, with the trend of decreasing weight gain

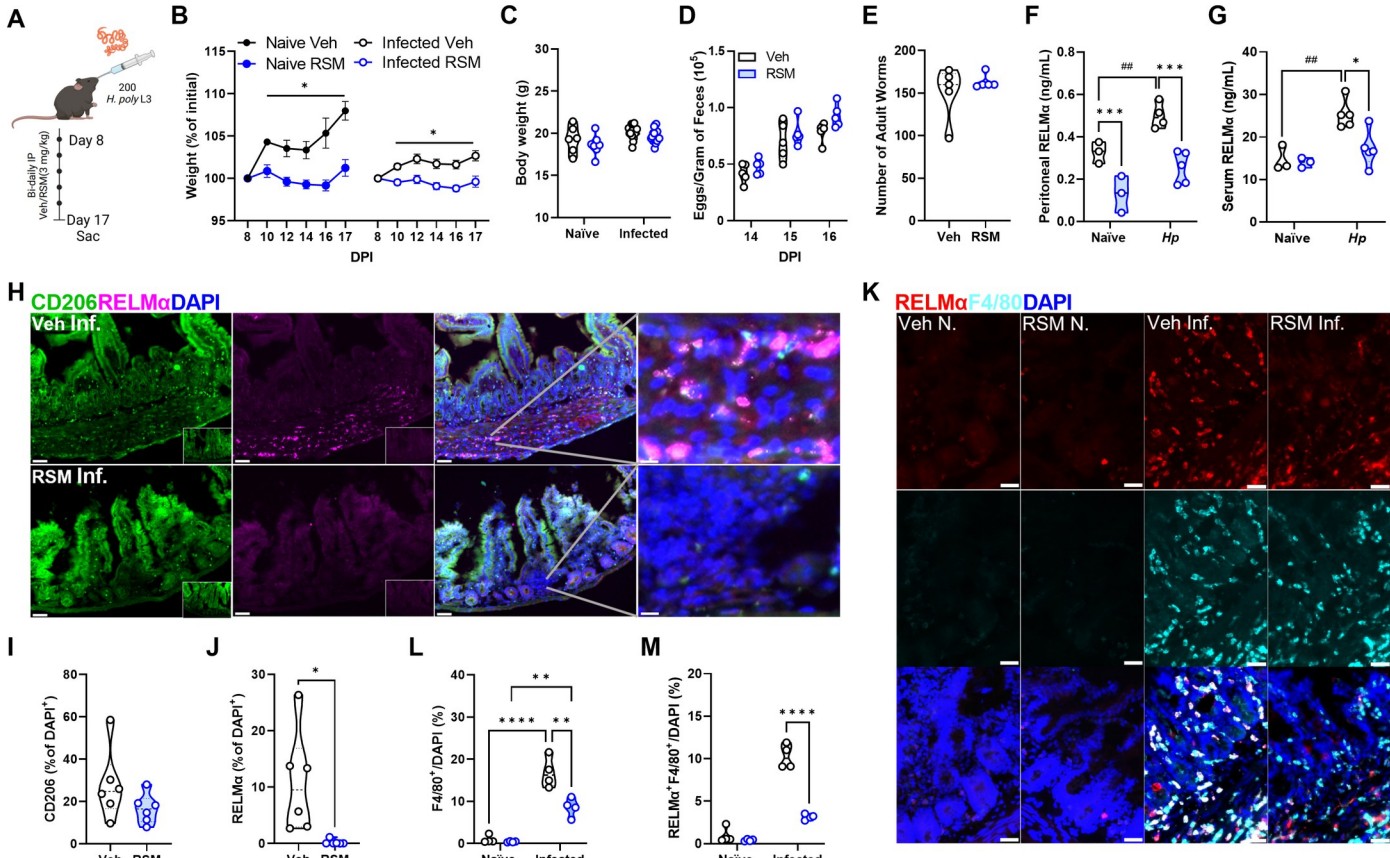

**Fig 7. *In vivo* choline kinase inhibition during primary infection impairs systemic and intestinal RELMα and intestinal macrophage alternative activation.** A) Schematic of primary infection with 200 *H. polygyrus* (*Hp*) L3 larvae through oral gavage. Mice were treated intraperitoneally with vehicle (40% DMSO in PBS) or RSM-932a (3 mg/kg) every other day for 8 days starting on day 8 and sacrificed on day 17. B-C) Percentage of weights at start of vehicle or RSM injections at 8 DPI with *H. polygyrus* (B) or final weights at day 17 (C). Mixed-effects analysis with Tukey's test for multiple comparisons (* $p < 0.05$). D-E) Eggs in feces were counted at multiple time points after infection, and E) adult worms were isolated from the small intestine and enumerated on the day of sacrifice. Values represent means ± SEM (n = 3–5 mice per group), representative of 3 experiments. Two-way ANOVA with Šídák's test for multiple comparisons and unpaired t test (ns). F-G) Detection of serum and G) peritoneal fluid RELMα by ELISA in naïve and *H. polygyrus*-infected mice. n = 3–5 per group. Two-way ANOVA with Šídák's test for multiple comparisons (* $p < 0.05$, *** $p < 0.001$ for differences between treatment and ## $p < 0.01$ for differences between naïve and infected vehicle-treated mice). H-L) Immunofluorescent staining of intestinal tissue for CD206 and RELMα against DAPI counterstain. Scale bar 50 μm. Quantification of CD206+ (I) or RELMα+ (J) per DAPI+ cell. Unpaired t test (* $p < 0.05$). Quantification of F4/80+ (K) and RELMα+F4/80+ (L) per DAPI+ cell. n = 6. Two-way ANOVA with Šídák's test for multiple comparisons (**** $p < 0.0001$). Schematics were created using BioRender.

but no change in final weights between groups (Fig 7B and 7C). IL-4-polarized macrophages have been shown to play an important role in reducing helminth parasite burden [36]; yet, neither egg nor worm burden were affected following RSM treatment (Fig 7D and 7E). *Hp* infection significantly increased RELMα levels in the peritoneal fluid and serum; nevertheless, consistent with *in vitro* and *in vivo* observations, RELMα levels were significantly decreased with RSM treatment (Fig 7F and 7G). Furthermore, the presence of CD206[+] and RELMα[+] cells in intestinal tissue, the primary site of infection, was also reduced (Fig 7H–7J). To determine whether macrophage derived RELMα was specifically affected by RSM treatment, intestinal tissue sections were co-stained with F4/80 and RELMα and double positive cells quantified. This analysis revealed that *Hp* infection strongly induced F4/80[+]RELMα[+] cells in the intestine, but these were significantly reduced by RSM treatment (Fig 7K and 7L).

We next probed the effect of chronic RSM treatment in a vaccination model for *Hp* involving secondary infection (S9A Fig), where IL-4-polarized macrophages are critical for protection [36]. Over the course of the 10-week infection, RSM treatment resulted in significantly increased fecal egg burden specifically at week 8 but showed a trend at all other time points (S9B Fig). Taking the area under the curve for egg burden, RSM treated mice had significantly higher total egg burden (S9C Fig), suggestive of some deficiency in optimal immunity to the parasitic worm. However, there were no significant differences in the adult worm burden after secondary challenge (S9D Fig). Remarkably, consistent with the acute regiment, RSM treatment led to a significant drop in peritoneal and serum RELMα (S9E and S9F Fig), as well as down-regulated CD206 and up-regulated CD86 expression in the intestinal cells of infected mice (S9G–S9I Fig). We conclude that choline metabolism inhibition significantly reduces RELMα but this is not sufficient to fully impact worm burden, however, can impact worm fertility. This may be partly explained by the fact that RSM does not inhibit other M2 macrophage effectors such as *Arg1*. Together, this data indicates the importance of choline metabolism and choline kinase signaling in promoting M[IL-4] polarization and RELMα expression in intestinal helminth infection.

## Discussion

Upon activation, macrophages up-regulate choline metabolism to fuel PC synthesis. LPS and IL-4 both induce expansion of organellar membranes including ER, mitochondria, and endo-lysosomes [37–40], in addition to a general enlargement of the cell itself [41]. Building on our understanding of macrophage choline metabolism during LPS polarization, we show here that stimulation of divergent signaling pathways (i.e., LPS-TLR4 vs. IL-4-JAK/STAT6), both lead to more choline taken into the cell with an associated up-regulation of Slc44a1, which likely facilitates increased flux through the PC synthetic pathway and more PC levels for membrane biogenesis. Recently, another putative choline transporter has been described, FLVCR1 [42]. *Flvcr1*-knockout mice are embryonic lethal, but little has been reported as to the role of this gene in immune cells. Interestingly, cells from *Flvcr1*-knockout embryos had major defects in mitochondrial structure and morphology. Our RNA-seq analysis revealed robust expression of *Flvcr1* in M[0] and M[IL-4], which was up-regulated by choline metabolism inhibition. In contrast, transcript expression of *Slc44a1* was inhibited by choline metabolism inhibition. The physiological role and individual contributions of choline transporters such as Slc44a1, Slc44a2 and Flvcr1 to choline transport and metabolism remains unexplored but of significant priority.

How phospholipid composition is affected by macrophage polarization has been addressed in studies using similar experimental designs. While the increased accumulation of PC in IL-4/IL13-polarized macrophages (RAW264.7 cells) was shown in a comprehensive assessment of

phospholipid content after long term exposure to polarizing stimuli, this was not seen in human THP-1-derived macrophages [43], nor were increases observed before 18 h [38]. Coupled with an increase in total PC content, we observed an increase in saturated and monounsaturated fatty acyl chains in PCs elevated in both M[IL-4] and M[LPS]. We and others have previously shown that M[LPS] up-regulate *de novo* lipogenesis to supply new fatty acids for phospholipid synthesis [4, 10]. There is now evidence to support a role for both exogenous fatty acid uptake [11, 12], as well as *de novo* lipogenesis in M[IL-4] [44]. Despite the potentially divergent sources of FA, differences in the side chain composition of choline-containing phospholipids between macrophage polarizations may fuel distinct lipid signaling pathways following phospholipase-specific processing.

A handful of studies have now interrogated the importance of choline metabolism in macrophage biology, though all have been in the context of LPS-induced polarization and inflammation [4, 5, 9, 45, 46]. Conversely, while macrophage IL-4 polarization is widely used *in vitro* and typically yields a reproducible response, how this might be affected by or might affect choline metabolism remained unclear. By blocking choline uptake and/or subsequent metabolism, the transcriptional induction of *Mrc1*, *Chil3*, *Arg1* were not changed. However, limiting the availability of choline in the media, or pharmacologically blocking the uptake/phosphorylation of choline caused the targeted reduction in IL-4-induced *Retnla* expression and RELMα secretion. Early studies indicated that signaling through STAT6 is essential for proper induction of Th2 cytokines and IL-4-induced RELMα expression [47]. In response to limiting choline availability and PC synthesis, STAT6 phosphorylation, as a surrogate for IL-4R signaling efficacy, was decreased. Though this supports the robust down-regulation of *Retnla*, similarly regulated transcripts such as *Arg1* and *Chil3* were not affected or mildly increased. Curiously, *Chil3* has even been suggested to augment *Retnla* expression [48]. The mechanism(s) by which choline availability regulates *Retnla* are likely transcriptional; however, the exact mechanism remains unclear. Interestingly, transient expression of *Retnla* was recently shown in nearly all tissue-resident macrophages and its induction in PEC macrophages was IL-4R- and STAT6-independent [49, 50]. Consistent with this, other studies report that RELMα is induced by signals other than Th2 cytokines, such as hypoxia or phagocytosis of apoptotic cells [51, 52]. Furthermore, a reporter mouse line showed RELMα expression mainly in WAT, intestine, and peritoneal macrophages [50], as well as peritoneal eosinophils and type II alveolar epithelial cells.

Our findings support the differential induction of canonical M[IL-4] genes and identify choline metabolism as a specific inducer of *Retnla*. In our study, we found that Chkα inhibition resulted in major changes to PEC populations, akin to the macrophage disappearance reaction upon inflammatory stimuli [53], pointing to a possible connection between macrophage tissue residence and choline metabolism. In addition, B cells in the peritoneum were drastically reduced, which corroborates previous observations demonstrating that RSM and another Chkα inhibitor (MN58B) dramatically lowered splenic B cell populations upon systemic delivery [54]. How this altered peritoneal immune profile may influence RELMα levels remains to be determined; however, since lung B cells express similar amounts of RELMα during *Nippostrongylus brasiliensis* infection as alveolar macrophages [55], the effect may not be exclusive to macrophages. During *Nb* infection, lung-resident interstitial macrophages up-regulate RELMα in a STAT6-dependent manner [50]. Separately, a subset of lung interstitial macrophages identified as CD169[+] nerve- and airway-associated macrophages (NAMs) were found to express high levels of RELMα [56]. Depletion of all interstitial macrophages impaired control of helminth infection [50], while depletion of NAMs exacerbated lung inflammation but had no impact on control of influenza infection [56].

Macrophage-derived RELMα has been shown to promote profibrotic collagen cross-linking in fibroblasts [57], while also contributing to pro-inflammatory processes in intestinal

inflammation and infection [33, 58, 59]. Given that limiting choline metabolism in M[IL-4] markedly depleted RELMα levels and that *in vivo* RSM treatment recapitulated this effect, we reasoned that blocking choline metabolism during helminth infection may phenocopy models of RELMα deficiency, where enhanced resistance to parasites is observed [60–62]. Contrary to our hypothesis, RSM-mediated choline kinase inhibition had minimal impact on indices of parasite burden in the helminth models we assessed, despite profound reductions in local and systemic levels of RELMα (Figs 6, S5, and S7). While the explanation for this remains unclear, there may be compensatory changes in RSM-treated mice, such as the altered immune profile, differential activation of macrophages, or direct effects of RSM on the pathogen that could confound the interpretation. The importance of macrophage-specific metabolism for M[IL-4] activation in the context of helminth infection has previously been shown. Inhibition of lysosomal lipolysis in *H. polygyrus*-infected mice reduced M[IL-4] polarization and RELMα expression, leading to increased parasite egg burden [63]. Similarly, our studies indicate that choline metabolism is also important in optimal M[IL-4] polarization in response to *H. polygyrus* infection, but with slight effects on egg burden in chronic studies and no effects on parasite burden. Future work could involve genetic and tissue-specific models to directly address the relevance of macrophage choline metabolism in controlling helminth infections.

Metabolic reprogramming that occurs in response to macrophage polarization represents one of the fundamental observations of immunometabolism. Our RNAseq analyses revealed intriguing changes in mitochondrial transcripts (ribosomal genes up-regulated and electron transport chain component transcripts down-regulated). The extent to which mitochondrial morphology and metabolism were altered was intriguing. M[IL-4] up-regulate oxidative and glycolytic programs, with key signals coming from metabolic intermediates such as α-ketoglutarate and succinate [64, 65]. When choline cannot enter the cell or be used in the synthesis of PC, cellular organelles are swiftly affected. The capacity to continuously provide PC, from ER to mitochondria is critically important and when disrupted, results in clear destruction of the mitochondrial cristae, results consistent with *Flvcr1-* and SLC44A1-deficient cells [5, 66]. As a result, macrophages, independent of IL-4 stimulation, must rely on anaerobic means to generate cellular ATP. Importantly, we observed a dramatic difference in the potency of inhibiting choline uptake via HC3 or choline kinase activity via RSM, whereby the latter caused the most dramatic effect. We reason that the inhibition of choline uptake, while limiting supply of exogenous choline, would continue to allow for the recycling and modification of existing phospholipids, which lessens the impact of the treatment. Choline kinase inhibition; however, would almost completely limit PC supply. While inhibiting macrophage PC synthesis was shown to lead to the loss of mitochondrial membrane potential, leading to mitophagy and a protection against inflammasome activation, future work will be required to identify whether metabolic reprogramming is mechanistically linked to *Retnla* down-regulation in M[IL-4] cells.

In addition to the striking effect on RELMα, we found the costimulatory checkpoint ligands PD-L1 and PD-L2 were coordinately regulated by choline metabolism, which is contrary to conventional IL-4-polarization. In tumour settings, nutrient availability may be heterogenous, especially for infiltrating immune cells, which can explain the wide range of PD-L1 positivity among tumour samples and the relatively poor response to anti-PD-1/PD-L1 immune-checkpoint inhibitor therapies [67]. Up-regulated choline metabolism is a recognized hallmark of certain cancers [68, 69] and targeting Chkα can both inhibit cancer cell growth and render cell death-resistant cancers more susceptible to immunotherapy [70]. Similarly, RSM-932A increased CD86 expression in the intestine in the secondary *H. polygyrus* infection, which is a hallmark of M1-polarized macrophages [71]. CD86 in antigen-presenting cells provides costimulatory signals to activate T cells through CD28 and CTLA4 binding [72, 73]. Previous

studies reported that CD86 was also up-regulated by helminth-derived peptide GK-1 in BMDM [74]. These changes in costimulatory and checkpoint proteins supports a potential immunoregulatory role of choline metabolism that may represent an underappreciated therapeutic potential in certain disease settings.

Macrophages up-regulate choline uptake and incorporation in response to IL-4 stimulation. Reciprocally, when the availability of choline is compromised, normal IL-4 responses are affected. We used two pharmacological approaches to limit choline availability and nutrient deprivation in the media. These approaches carry numerous caveats, including dose, timing, route, and potential off-target effects; however, our *in vitro* pharmacological approach targeted distinct steps in the CDP-choline pathway. Separately, other fates of choline may be modified by or play a role in mediating the alteration of immune cell function. For instance, phosphocholine may post-translationally modify proteins [75, 76], and macrophages and other immune cells have been shown to produce acetylcholine [77]. Conversion of choline to betaine may also support osmoregulation [78] or contribute to histone methylation [79], but these latter mechanisms require deeper investigation in immune contexts and remain speculative. Future work using genetic models to solidify the importance of choline uptake and subsequent metabolism are warranted.

In summary, we describe a critical role for choline metabolism in the mediating the full potential of macrophage M[IL-4] polarization and metabolic programming. Moreover, systemic pharmacological inhibition of choline metabolism in mice closely mirrors the dramatic down-regulation of RELMα but fails to affect the pathology of intestinal or lung helminth infection models. This work also highlights that the inhibition of choline metabolism via RSM caused a dramatic shift in immune cell profile and potentially polarization, which may be therapeutically beneficial in other disease settings such as peritoneal metastases or fibrosis.

## Methods

### Ethics statement

All animal procedures were approved by the University of Ottawa Animal Care Committee (BMI-1863) and the University of California Riverside Institutional Animal Care and Use Committee (protocol A-20210017).

### Animals

Mice (C57BL/6J) were originally purchased from Jackson Laboratories (Stock no. 00064) and bred in a pathogen-free facility in the University of Ottawa animal facility or acclimated for 7 days prior to *H. polygyrus* infection at the University of California Riverside. Mice were maintained on a 12-hour light dark cycle (lights on at 7:00 am) and housed at 23°C with bedding enrichment. Male and female mice ages 8–16 weeks were used for the generation of primary macrophages as described below.

### Isolation, culturing and polarization of bone marrow-derived macrophages

BMDM were isolated and cultured as previously described [80, 81]. Briefly, bone marrow cells were obtained from the femur and tibia by centrifugation [4]. Cells were differentiated into macrophages using 15–20% L929-conditioned media in complete DMEM (Wisent) containing 10% FBS (Wisent), 1% penicillin/streptomycin. Cells were plated into 15 cm dishes and allowed to differentiate for 6–8 days. Cells were lifted by gentle scraping in 10mM EDTA in PBS, counted, and seeded into culture plates for experiments at $1\times10^6$/mL. Cells were treated for 24 h with DMSO as vehicle or inhibitors: hemicholinium-3 (Sigma-

Aldrich), RSM-932a (Cayman Chemicals). Macrophages were polarized with 20 ng/mL recombinant IL-4 (Peprotech or Roche) or 100 ng/mL LPS (*E. coli*:B4, Sigma-Aldrich). Choline-free DMEM was formulated by preparing nutrient-deficient DMEM (US Biologicals, D9809) according to manufacturer's instructions and supplementing with sodium pyruvate (Gibco), *myo*-inositol, L-methionine, and calcium D-pantothenate (Sigma-Aldrich).

## Choline uptake, incorporation, uptake inhibition and degradation

The rates of choline uptake, incorporation into PC, uptake inhibition and degradation were determined in M[0] and M[IL-4] using [3]H-choline chloride (Perkin Elmer) as previously described [4].

## High-performance liquid chromatography electrospray ionization tandem mass spectrometry (LC-ESI-MS/MS) Lipidomics

Cells were counted, pelleted, and frozen at -80˚C in 15 mL Falcon tubes for 24 h. Pellets were resuspended in 1 mL sodium acetate using a syringe pipette and transferred to Kimble tubes. Tubes were washed two times in 1 mL and 1.2 mL sodium acetate, respectively, to recover all material and these volumes were also transferred to the corresponding Kimble tube. All pipette tips were glass. MS-grade lipid standards (all from Avanti Polar Lipids) consisting of 90.7 ng LPC (13:0/0:0), 100 ng PC(12:0/13:0), and 75 ng SM(d18:1/18:1-D9), were added to the samples at time of extraction. Lipids were extracted using a modified Bligh and Dyer [82] protocol at a final ratio of acidified methanol/chloroform/sodium acetate of 2:1.9:1.6 as previously described [83, 84]. The organic chloroform phase was retained, and the aqueous phase was successively back-extracted using chloroform three times. The four chloroform extracts were combined and evaporated at room temperature under a constant stream of nitrogen gas. Final extracts were solubilized in 300 μL of anhydrous ethanol (Commercial Alcohols) and stored under nitrogen gas at -80˚C in amber vials (BioLynx).

Lipid quantification was performed on an Agilent 1290 Infinity II liquid chromatography system coupled to a QTRAP 5500 triple quadrupole-linear ion trap mass spectrometer using a Turbo V ion source (AB SCIEX) in positive ion mode. Reverse phase chromatography was performed using a binary solvent system composed of solvent A (water with 0.1% formic acid (Fluka) and 10 mM ammonium acetate (OmniPur)) and solvent B (acetonitrile (J.T. Baker) and isopropanol (Fisher) at a ratio of 5:2 v/v with 10 mM ammonium acetate and 0.1% formic acid) and a nanobore 100 mm x 250 μm (i.d.) capillary column packed in-house with Repro-Sil-Pur 200 C18 (particle size of 3 μm and pore size of 200 Å; Dr. A. Maisch, Ammerbruch). Three μL of sample at 4˚C was injected with the system operating at 10 μL/min and using a gradient that started at 30% solvent B, reached 100% solvent B at 8 minutes, and remained at 100% solvent B for 45 min. At 45 min, composition was returned to 30% solvent B and the column was regenerated for 15 min. Samples were prepared for HPLC injection by mixing 5 μL of lipid extract with 2.5 μL of an internal standard mixture (all from Cayman Chemicals) consisting of 2.5 ng LPC(O-16:0-D4/0:0), 2.5 ng LPC(O-18:0-D4/0:0), 1.25 ng PC(O-16:0-D4/2:0) and 1.25 ng PC(O-18:0-D4/2:0) in EtOH, and 16 μL of Solvent A. Transition lists were established using precursor ion scan interrogating a diagnostic fragment ion at m/z 184.1, corresponding the phosphorylcholine headgroup of the protonated molecular ion ([M+H][+]). Once all detectable species were established in precursor ion discovery mode, targeted lipid quantification was performed using selected reaction monitoring (SRM), monitoring transitions from [M+H][+] with the product ion 184.1. ESI-MS/MS acquisition and instrument control were performed using Analyst software (version 1.6.3, SCIEX). The ion source operated at

5500 V and 0˚C. Nebulizer/heated gas (GS1/GS2), curtain gas, and collision gas (all nitrogen) were set to 20/0 psi, 20 psi, and medium, respectively. Compound parameters (de-clustering potential, entrance potential, collision energy, and collision cell exit potential) were individually optimized for each transition. MultiQuant software (version 3.0.2 AB SCIEX) was used for peak picking and processing quantitative SRM data. Bayesian Annotations for Targeted Lipidomics (BATL v2.7, https://www.complimet.ca/shiny/batl/) was used to assign peaks [84]. Peak areas were normalized to both cell number and either LPC(13:0/0:0) or SM(d18:1/18:1-D9) for PCs and SMs, respectively to account for extraction efficiency and instrument response. Data are expressed as nmol or pmol equivalents per $10^6$ cells.

The identities of all lipid species in our samples were structurally determined from quality control samples composed of an equi-volume pool of all samples analyzed. SRM was used as a survey scan to trigger information dependent acquisition of EPI spectra in the linear ion trap. After acquisition, the EPI fragment spectra were analyzed to determine structural identities of the lipids.

## Wound healing

3T3-L1 fibroblasts were passaged in complete DMEM, plated in ImageLock plates (Essen Bioscience) and grown to confluency. Macrophages were pre-treated with vehicle or HC3 for 24 h, washed with PBS, and supernatants were collected after a further 24 h in complete DMEM ± 20 ng/mL IL-4. A scratch wound was made in each well of 3T3-L1 fibroblasts, wells were gently washed with warm PBS, and supernatants were added to 4 or 5 replicate scratch wound wells. Wound healing was monitored hourly for >24 h on an IncuCyte ZOOM system (Essen Bioscience).

## Flow cytometry

Macrophages were plated in 6-well plates and treated with DMSO or HC3 (250 μM) for 48 h. After washing with PBS, cells were gently scraped or lifted with 10 mM EDTA in PBS. For naïve *in vivo* experiments, peritoneal exudate cells (PECs) were recovered in a total of 6–8 mL of ice-cold PBS. For collection of peritoneal cellular aggregates, the inflated peritoneal cavity was vigorously massaged and cut open over a funnel [31]. Peritoneal aggregates were allowed to settle out of the lavage fluid for at least 20 min on ice and suspended PECs were aspirated with a plastic Pasteur pipette. Peritoneal aggregates and dissected omenta were digested in 0.25 mg/mL collagenase IV (Sigma) and 0.25 mg/mL DNase I (ThermoFisher) in RPMI-1640 (Wisent) with 5 mM HEPES for 15 min at 37˚C. Cell suspensions were blocked and stained with anti-CD16/32 (93, BioLegend) and Zombie Aqua dye (BioLegend) in PBS for 30 minutes on ice. Surface staining in PBA-E (1% BSA, 2 mM EDTA, 0.05% NaN₃ in PBS) was done for 20 minutes on ice. Cells were then fixed with 2% paraformaldehyde in PBS or Fix/Perm buffer (ThermoFisher) for 15 minutes. Intracellular staining was performed by permeabilization for 5 minutes on ice with 0.5% saponin in PBA-E (PBA-S) or permeabilization buffer (Thermo-Fisher) and subsequent staining with antibodies diluted in PBA-S. For intracellular phospho-protein staining, cells were fixed directly after harvesting with -20˚C phospho-flow fix buffer (4% paraformaldehyde/90% methanol in PBS). Surface antibodies were purchased from BioLe-gend or ThermoFisher: F4/80-PE (BM8), F4/80-AF488 (BM8), F4/80-PE-Dazzle594 (BM8), PD-L1-BV421 (10F.9G2), PD-L2-PE (TY25), RELMα-PerCP-eF710 (DS8RELM), Egr2-APC (erongr2), CD38-eF450 (90), pSTAT6(Tyr641)-APC (CHI2S4N), CD11b-APC-eF780 (M1/70), Ly-6C-PE-Cy7 (HK1.4), Ly-6G-AF647 (1A8), Ly-6G-FITC (1A8), CD19-Pacific Blue (6D5), SiglecF-SB600 (1RMN44N), CD86-FITC (GL1), MHCII-PE (M5/114.15.2). Unconjugated polyclonal rabbit anti-RELMα (Peprotech) was detected with goat anti-rabbit AF647

plus (ThermoFisher). Cells were washed and resuspended in PBA-E and acquired using LSR Fortessa or FACSCelesta flow cytometers (BD) and analyzed using FlowJo v10.7.2 (BD).

For *in vivo* helminth experiments, PECs were recovered in a total of 5 mL of ice-cold PBS. Cells were blocked with 0.6 μg Rat IgG and 0.6 μg anti-CD16/32 (93) for 5 minutes and stained for 30 minutes with antibodies to Ly6G (1A8), CD3e (145-2C11), PD-L1 (MIH6), CD86 (GL-1), CD4 (RM4-5), MERTK (2B10C42), CD301b (LOM-14), PD-L2 (TY25), CD5 (53–7.3), Ly6C (HK1.4), MHCII (M5/114.15.2) (BioLegend, San Diego, CA); CD8 (3B5), CD11b (M1/70) (Invitrogen); SiglecF (E50-2440) (BD Bioscience); CD206 (MR5D3) (BioRad). Cells were then washed and analyzed on a Novocyte (ACEA Biosciences) followed by data analysis using FlowJo v10.8.1 (BD). Frequencies or geometric mean fluorescence intensity (gMFI) were calculated.

## Transcript and protein expression

Total RNA was isolated from BMDM using the TriPure reagent protocol (Roche Life Sciences). Isolated RNA was re-suspended in RNase/DNase-free water (Wisent). QuantiNova™ reverse transcription kit (Qiagen) or ABM All-in-one RT kit (ABM) was used to synthesize cDNA according to manufacturer's instructions. To determine transcript expression, the Quanti-Nova™ Probe PCR kit (Qiagen) was used in combination with hydrolysis primer-probe sets (ThermoFisher) or BrightGreen Express MasterMix (ABM) with custom-designed primers (ThermoFisher). Comparative qPCR reactions were run on the Roto-Gene Q (Qiagen) or Stratagene MX3005p (Agilent). Relative transcript expression was determined using the delta-delta $C_q$ method and normalized to *Actb* [85]. Following treatments, macrophage protein expression and phosphorylation status were assessed by immunoblotting as previously described [4].

For RNA-seq analysis, total RNA was isolated from BMDM using Tripure reagent and chloroform and subsequently purified using spin columns (PureLink RNA Mini, ThermoFisher) according to manufacturer instructions. Messenger RNA was purified from total RNA using poly-T oligo-attached magnetic beads. After fragmentation, the first strand cDNA was synthesized using random hexamer primers followed by the second strand cDNA synthesis. The library was ready after end repair, A-tailing, adapter ligation, size selection, amplification, and purification. The minimum Q30 score was 90.96%.

## Dextran sodium sulfate (DSS)-induced colitis

Mice were provided 2–2.5% dextran sodium sulfate (MW 40–50 kDa, MP Biomedical) in drinking water *ad libitum* and treated intraperitoneally with vehicle (40% DMSO in PBS) or RSM-932a (3 mg/kg) every other day for 6 days and sacrificed at day 7. Colons were dissected out and measured with uniform tension by hanging the caecum from a clamp and applying a weight to the distal end. Liver, white adipose tissue, and 0.5-cm pieces of colon were snap-frozen in liquid $N_2$ or fixed in 10% neutral-buffered formalin (Sigma-Aldrich). Feces was collected daily and hemoccult was assessed by luminescence with ECL substrate (BioRad).

## RELMα ELISA

Recombinant RELMα, polyclonal rabbit anti-RELMα capture, and polyclonal biotinylated rabbit anti-RELMα detection antibodies (all Peprotech) were used according to a previously described protocol [62].

## Heligmosomoides polygyrus (Hp) *infection model*

*Hp* life cycle was maintained in mice, as previously described [86, 87]. Mice were orally gavaged with 200 *Hp* L3 stage larvae or with PBS (naive group). Egg burden was measured at

indicated day post-infection by counting eggs in feces with McMaster slide. To enumerate L5 stage *Hp*, the small intestines of infected mice were cut longitudinally, and larvae were isolated by fine forceps. Isolated larvae were counted on the petri dish with grids under the dissection microscope. *Hp* infection experiments were conducted at least twice and as indicated in the figure legends.

## RSM-932A

RSM-932A (Cayman Chemical, Ann Arbor, MI) was reconstituted to 30 mg/mL in DMSO. For *in vivo* injection, RSM-932A was further diluted with DMSO and PBS to 1.5 mg/mL. Mice were injected intraperitoneally with 3 mg/kg of RSM-932A or 40% DMSO in PBS (vehicle group). *In vivo* injection experiments were conducted at least 3 independent times and as indicated in the figure legends.

## RNA-seq data analysis

Raw paired-end reads were trimmed of adapters using fastp [88] before being aligned on the mouse genome (assembly GRCm38) using STAR [89] in local mode and retaining only read pairs with mapq score of 40. Duplicate reads were marked but retained using Picard [90]. Reads aligning to exons were counted and summarized to genes using Subread featureCounts [91, 92]. Only genes with at least 10 counts in at least two samples were retained for further analysis. Differential expression testing was performed in R/Bioconductor [93] using edgeR and the glmQLFTest function [94]. For differential expression calls, a log2 fold-change value of 1 or -1 combined with a Benjamini-Hochberg-calculated FDR value less than 0.05 was judged significant [95]. Differentially expressed gene sets (up- or down-regulated genes with log2 fold-change of at least 1 or -1 and FDR lower than 0.05) were searched for potential enrichment in gene functional terms. The R/Bioconductor packages ReactomePA [96] and clusterProfiler [97] were used to search for enrichment in Reactome terms [98]. In these analyses, the search universe was limited to genes filtered for minimum expression.

## Extracellular flux analysis

BMDM were plated in 96-well culture plates (Agilent) at $7.5 \times 10^4$ cells/well in complete DMEM. Cells were treated with DMSO, HC3, or RSM-932a at different concentrations ± 20 ng/mL IL-4 for 24 h. Extracellular flux analysis was performed on XFe96 (Agilent) using an adapted MitoStress Test Kit (Agilent). Cartridge ports were loaded with 10x concentrations of drugs: 15 μM oligomycin, 140 μM BAM15 (Cayman Chemicals), and 10μM rotenone/10 μM antimycin A/20 μM Hoechst 33342. BAM15 is a mitochondrial uncoupler with improved induction of maximal respiration compared to FCCP [99]. RSM932a was loaded in place of rotenone/antimycin A in some wells. Data were normalized by cell counts obtained by nuclear Hoechst 33342 quantification on an EVOS FL Auto 2 (ThermoFisher) microscope and Qupath (University of Edinburgh). Bioenergetics were calculated using a worksheet template [26].

## Transmission electron microscopy

BMDM were plated in 6-well plates and treated with DMSO or RSM-932a ± 20 ng/mL IL-4 for 24 h. Cells were lifted by gentle scraping in 10mM EDTA in PBS and fixed in 4% paraformaldehyde/3.5% glutaraldehyde overnight. Cell pellets were embedded in agarose and blocks were sectioned at 50-μm thickness using a Lecia VT1000S vibratome. Resin blocks were sectioned at 80-nm thickness using a Leica EM UC6 ultramicrotome and placed on microgrids. Sections

were imaged on a JEM-1400plus instrument (JEOL) at 80 kEV and 5600x or 27600x magnification.

## Immunofluorescence staining

Intestinal tissue was stored overnight in 4% PFA at 4°C. After 24 hours, tissue was removed from 4% PFA and incubated for 24 hours in 30% sucrose. Intestines were blocked in OCT and sectioned at 10 μm. Tissue sections were incubated with polyclonal rabbit anti-CD86 (ThermoFisher), RELMα (Peprotech), APC-conjugated anti-RELMα (DS8RELM; Invitrogen), anti-F4/80 (CI:A3-1), and biotinylated anti-CD206 (C068C2; BioLegend) antibodies overnight at 4°C. Tissue sections were then incubated with Cy2-conjugated streptavidin (Jackson ImmunoResearch), Cy3-conjugated goat anti-rabbit antibodies (Abcam), TRITC-conjugated chicken anti-rabbit IgG (Invitrogen), or Cy5-conjugated donkey anti-rat IgG (Life Technologies) for 2 hours at 4°C and mounted with VECTASHIELD HardSet Anti-fade Mounting Medium (Vector Laboratories) followed by imaging with BZ-X800 microscope (Keyence). Positive cells were counted by QuPath 0.3.2 (University of Edinburgh) [100].

## Statistical analysis

Statistical analysis was performed in GraphPad Prism version 9.5.1 for Windows (GraphPad Software). The number of times individual tests were replicated, and biological sample sizes (number of mice) are described in figure legends.

## Supporting information

**S1 Fig. Choline-containing phospholipid species in polarized macrophages.** A-I) Sum of phospholipid subclasses. A) Plasmenyl-lysophosphatidylcholines (LPC(P)), B) plasmanyl-LPCs (LPC(O)), C) platelet activating factors (PAF). D) D) Heatmap of sphingomyelin (SM), plasmanyl-PAFs (PC(O)-PAF), PC(O), PC(P), LPC, LPC(O), and LPC(P). Heatmap statistics are shown as fold change over the average of M[0]. n = 5. One-way ANOVA with Dunnett's test vs M[0] for multiple comparisons ($^*$ p < 0.05, $^{**}$ p < 0.01, $^{****}$ p < 0.0001).
(TIF)

**S2 Fig. IL-4 up-regulates choline metabolism in macrophages.** A) Saturation curves showing rate of $^3$H-choline uptake at increasing concentrations of unlabeled choline. n = 8. Michaelis-Menten least squares fit regression ($^{****}$ p < 0.0001). B) Inhibition of $^3$H-choline uptake by HC3. n = 4 (M[0]) or 5 (M[IL-4]). Four parameter log(inhibitor) vs. response regression F test ($^{**}$ p < 0.01). C) Pulse-chase to determine the rate of PC degradation. n = 3. Linear regression with F test for slopes (ns).
(TIF)

**S3 Fig. Choline uptake and phosphorylation are required for normal IL-4 signaling and M [IL-4] phenotype.** A-C) Macrophages were treated with vehicle (DMSO) or RSM-932a (5 μM) for 24 h, washed, then treated with IL-4 (20 ng/mL) for 24 h. Relative expression of M[IL-4] hallmark genes *Retnla*, *Arg1*, *Mrc1*, normalized to *Actb* and compared to M[0]. n = 3–4, representative of 3 experiments. Unpaired t test ($^{**}$ p < 0.01). D-E) Macrophages were treated with vehicle (DMSO) or HC3 (250 μM) for 6 h, washed, then treated with IL-4 (20 ng/mL) for 24 h. Relative expression of *Retnla* or *Il4ra* (E) normalized to *Actb* and compared to M[0]. n = 3. F) Macrophages were treated with vehicle (DMSO) or RSM-932a (5 μM) for 24 h, washed, then treated with IL-4 (20 ng/mL). Expression of IL-4 signaling molecule pSTAT6 (Tyr641) compared to β-actin. Representative of n = 5. G) Macrophages were treated with vehicle (DMSO)

or HC3 (250 μM) for 24 h, washed, then treated with IL-4 (20 ng/mL) for 24 h. Expression of intracellular Egr2 or surface CD38. n = 3, representative of 2 experiments. H) Left, schematic of wound healing assay with conditioned media. Right, Macrophages were treated with vehicle (DMSO) or HC3 (250 μM) for 24 h, washed, then treated with IL-4 (20 ng/mL) for 24 h and conditioned media was collected. Confluent 3T3-L1 fibroblast monolayers were scratched, and media was replaced with macrophage conditioned media. Images of wound healing over time, quantified by wound density. n = 3, representative of 2 experiments. Sum-of-squares F test of non-linear fit of growth curves (**** $p < 0.0001$). Lower, snapshots of 3T3-L1 fibroblast wounds at different timepoints. Schematics were created using BioRender.
(TIF)

**S4 Fig. Reactome pathway analysis.** A-D) Reactome pathway analysis of genes up-regulated by HC3 (A) or RSM (C) in M[IL-4] or down-regulated by HC3 (B) or RSM (D), sorted by gene count enrichment for each Reactome pathway.
(TIF)

**S5 Fig. Mitochondrial changes induced by inhibiting choline metabolism.** A) Densitometry of complex III and I from Fig 4B. Two-way ANOVA with Dunnett's test for multiple comparisons. B) Bioenergetics analysis of ATP produced through oxidative phosphorylation (ATP$_{OXPHOS}$) or glycolysis (ATP$_{Glyco}$). Macrophages were treated for 1 h or 5 min with RSM (0.2, 1, or 5 μM) prior to Mito Stress Test assay as in Fig 4C–4F. Measurements (n = 3 in triplicate) were normalized per $10^3$ cells. Mixed-effects analysis with Tukey's test for multiple comparisons (ATP$_{OXPHOS}$: # $p < 0.05$, ## $p < 0.01$, ### $p < 0.001$; ATP$_{Glyco}$: ** $p < 0.01$). C) Mito Stress Test assay of extracellular flux with sequential treatments of 1.5 μM oligomycin, 14 μM BAM15, and 1 μM rotenone/1 μM antimycin A/Hoechst 33342. Oxygen consumption rate (OCR; B) or extracellular acidification rate (ECAR; C) of M[0] or M[IL-4] treated with RSM (1 or 5 μM) in place of rotenone/antimycin A during the last injection. D) Histograms and geometric MFI of tetramethylrhodamine methyl ester (TMRM) staining of macrophages treated with inhibitors or vehicle (DMSO).
(TIF)

**S6 Fig. *In vivo* consequence of inhibiting choline metabolism.** A) Gating strategies for peritoneal cells. B) Schematic of 3-day *in vivo* choline kinase inhibition. Mice were treated intraperitoneally with vehicle (40% DMSO in PBS) or RSM-932a (3 mg/kg) on day 0 and 2 and sacrificed on day 3. n = 3–4. C) Intracellular RELMα expression in live CD11b$^+$ F480$^{hi}$MHCII$^{lo}$, F480$^{lo}$MHCII$^+$, F480$^-$, Ly6G$^+$ PMN, and CD11b$^{lo}$ peritoneal cells. Two-way ANOVA with Šídák's test for multiple comparisons (*** $p < 0.001$). D-G) Expression of *Emr1* (*Adgre1/F4/80*), *Retnla*, *Chil3*, or *Mrc1* in WAT. Unpaired t test (* $p < 0.05$, ** $p < 0.01$). H) Schematic of DSS-induced colitis. Mice were given 2–2.5% DSS in drinking water and treated with vehicle (40% DMSO in PBS) or RSM-932a (3 mg/kg) every other day for 6 days and sacrificed on day 7. I) Body weight was measured daily. n = 8–9, representing 2 independent experiments. J) Colon length on day 7. Reference naïve colon length in dashed line (8.025cm) from n = 4 mice. Unpaired t test (*** $p < 0.001$). K) Hemoccult detected in feces collected on day 6 normalized to weight. n = 4–5, representative of 2 experiments. Mann Whitney U test (* $p < 0.05$). Schematics were created using BioRender.
(TIF)

**S7 Fig. Expression of activation markers in peritoneal macrophages during helminth infection.** A) Gating strategy of peritoneal cell populations. B) Enumeration of MHCII$^{hi}$ and MCHII$^{lo}$ large and small peritoneal macrophages among live PECs. n = 4–5, representative of 2 experiments. Unpaired t test (** $p < 0.01$, **** $p < 0.0001$). C-F) CD86, D) CD206, E)

PD-L, or F) PD-L2 in MHCII^hi and MCHII^lo large and small peritoneal macrophages. Unpaired t test (*, $p < 0.05$, ** $p < 0.01$, *** $p < 0.001$, **** $p < 0.0001$).
(TIF)

**S8 Fig. *In vivo* choline kinase inhibition during secondary intestinal helminth infection alters peritoneal cell populations and macrophage alternative activation.** A-D) Enumeration of A) B-1 cells, B) monocytes, C) neutrophils, D) peritoneal macrophages among live PECs. n = 4–5. Unpaired t test (** $p < 0.01$, **** $p < 0.0001$). E-F) CD86, or G) CD206 expression (gMFI) on peritoneal macrophages. n = 4–5, representative of 2 experiments. Unpaired t test (**** $p < 0.0001$).
(TIF)

**S9 Fig. Long-term Chkα inhibition in a chronic model of *H. polygyrus*.** A) Schematic of secondary infection. Mice were infected with 200 *H. polygyrus* L3 larvae through oral gavage and intraperitoneally injected with vehicle (40% DMSO in PBS) or RSM-932a (3 mg/kg) every other day from day 2. Mice were treated with pyrantel pamoate (1 mg) at day 25, challenged with 200 *H. polygyrus* L3 larvae at day 42, and sacrificed at day 52 (D-G). For a long-term chronic infection (C), mice were infected with 200 *H. polygyrus* L3 larvae and intraperitoneally injected with vehicle (40% DMSO in PBS) or RSM-932a (3 mg/kg) two or three times per week. Then, mice were treated with pyrantel pamoate (1 mg) at day 93 and challenged with 200 *H. polygyrus* L3 larvae at day 107 and sacrificed at day 117. B-D) Eggs in feces were counted at multiple time points after long-term chronic infection, and the area under the curve (AUC) calculated (C). D) adult worms were isolated from the small intestine and enumerated at 52 DPI. n = 4–5 mice, representative of 2 experiments. Two-way ANOVA with Šídák's test for multiple comparisons or unpaired t test (* $p < 0.05$). E-F) Detection of peritoneal fluid and F) serum RELMα by ELISA in naïve and *H. polygyrus*-infected mice. n = 3–5 per group, representative of 2 experiments. Unpaired t test (** $p < 0.01$, **** $p < 0.0001$). G-I) Representative immunofluorescent images of intestinal tissue stained for CD86, CD206, and RELMα against DAPI counterstain. Scale bar 50 μm. Quantification of CD206^+ (H) or CD86^+ (I) per DAPI^+ cell. n = 5, representative of 2 experiments. Unpaired t test (** $p < 0.01$). Schematics were created using BioRender.
(TIF)

**S10 Fig. Graphical abstract.** Created using BioRender.
(TIF)

**S1 Data. Raw data file.**
(XLSX)

# Acknowledgments

We thank Dr. Vera Tang and the Faculty of Medicine Flow Cytometry Core Facility (RRID: SCR_023306), Dr. Chloë van Oostende-Triplet and the Cell Biology and Image Acquisition Core Facility (RRID: SCR_021845), the Faculty of Medicine Transmission Electron Microscopy Core Facility, as well as the Animal Care and Veterinary Services at the University of Ottawa, all of whom receive funding from the University of Ottawa and the Canadian Foundation for Innovation's Infrastructure Operating Fund. This work benefitted from data assembled by the ImmGen consortium. We thank the Digital Alliance of Canada for access to the Cedar high-performance computing cluster. The authors would like to apologize to the colleagues whose significant work could not be included due to length, citation limitations, or author oversight.

## Author Contributions

**Conceptualization:** Peyman Ghorbani, Sang Yong Kim, Meera G. Nair, Morgan D. Fullerton.

**Formal analysis:** Peyman Ghorbani, Sang Yong Kim, Irina Alecu, Steffany A. L. Bennett, Baptiste Lacoste, Alexandre Blais, Meera G. Nair, Morgan D. Fullerton.

**Funding acquisition:** Steffany A. L. Bennett, Alexandre Blais, Meera G. Nair, Morgan D. Fullerton.

**Investigation:** Peyman Ghorbani, Sang Yong Kim, Tyler K. T. Smith, Lucía Minarrieta, Victoria Robert-Gostlin, Marisa K. Kilgour, Maja Ilijevska, Irina Alecu, Shayne A. Snider, Kaitlyn D. Margison, Julia R. C. Nunes, Daniel Woo, Ciara Pember, Conor O'Dwyer, Julie Ouellette, Pavel Kotchetkov, Baptiste Lacoste.

**Resources:** Julie St-Pierre.

**Supervision:** Julie St-Pierre, Steffany A. L. Bennett, Baptiste Lacoste, Meera G. Nair, Morgan D. Fullerton.

**Visualization:** Alexandre Blais.

**Writing – original draft:** Peyman Ghorbani, Sang Yong Kim, Meera G. Nair, Morgan D. Fullerton.

**Writing – review & editing:** Peyman Ghorbani, Sang Yong Kim, Tyler K. T. Smith, Lucía Minarrieta, Marisa K. Kilgour, Maja Ilijevska, Irina Alecu, Shayne A. Snider, Kaitlyn D. Margison, Julia R. C. Nunes, Daniel Woo, Ciara Pember, Conor O'Dwyer, Julie St-Pierre, Steffany A. L. Bennett, Meera G. Nair, Morgan D. Fullerton.

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
