## [Decision Letter · Decision Letter 0]

6 Feb 2023

Dear Dr Fullerton,

Thank you very much for submitting your manuscript "Choline metabolism underpins macrophage IL-4 polarization and RELMα up-regulation in helminth infection" for consideration at PLOS Pathogens. As with all papers reviewed by the journal, your manuscript was reviewed by members of the editorial board and by several independent reviewers. In light of the reviews (below this email), we would like to invite the resubmission of a significantly-revised version that takes into account the reviewers' comments.

The reviewers found these data intriguing, but there are significant weaknesses that were noted. Please address the points of all 3 reviewers, particularly the disappearance on peritoneal macs, the disappearance of multiple cells in the treated infected animals, the specificity of the inhibitor (need a genetic approach), full phenotyping of the macrophages in the absence of Relma. Additionally the discussion needs to properly place the data in the context of the field.

We cannot make any decision about publication until we have seen the revised manuscript and your response to the reviewers' comments. Your revised manuscript is also likely to be sent to reviewers for further evaluation.

Sincerely,

Keke C Fairfax, PhD

Guest Editor

PLOS Pathogens

P'ng Loke

Section Editor

PLOS Pathogens

Kasturi Haldar

Editor-in-Chief

PLOS Pathogens

orcid.org/0000-0001-5065-158X

Michael Malim

Editor-in-Chief

PLOS Pathogens

orcid.org/0000-0002-7699-2064

The reviewers found these data intriguing, but there are significant weaknesses that were noted. Please address the points of all 3 reviewers, particularly the disappearance on peritoneal macs, the disappearance of multiple cells in the treated infected animals, the specificity of the inhibitor (need a genetic approach), full phenotyping of the macrophages in the absence of Relma. Additionally the discussion needs to properly place the data in the context of the field.

Reviewer's Responses to Questions

**Part I - Summary**

Reviewer #1: The manuscript by Ghorbani and Kim et al. shows an interesting role for choline metabolism in macrophage polarization, phenotype, or function in response to IL-4. IL-4 exposure increased choline metabolism and altered lipidome composition in macrophages. In vitro, pre-treatment of macrophages with inhibitors of choline metabolism resulted in lower expression of the M[IL-4] product RELMa and lower pSTAT6 levels in response to IL-4. In vivo, inhibition of choline metabolism led to alterations in cellular composition and RELMa expression in macrophages in the peritoneal cavity. During primary and secondary helminth infection with Heligmosomoides polygyrus and primary infection with Nippostrongylus brasiliensis, choline metabolism inhibition led to decreased RELMa levels without impacting parasite burdens or egg production.

Reviewer #2: In the present manuscript, Ghorbani et al. show that choline uptake and metabolism are altered in IL-4-polarized bone-marrow derived macrophages (BMDMs), leading to increased intracellular phosphatidylcholine (PC) levels. Using pharmacological inhibitors of the choline metabolic pathway, they mainly report alterations of macrophage polarization - mostly based on RELMalpha expression - in response to IL-4 and helminth infection. Altogether, the manuscript contains some interesting findings but derived exclusively from studies performed with pharmacological inhibitors with doubtful specificity, especially in vivo. In addition to complementary genetic approach(es), some mechanistic aspects would also need to be clarified to strengthen the manuscript and the authors’ conclusions.

Reviewer #3: Choline is an essential nutrient that forms the main phospholipid class in immune cells. Recent evidence suggests different inflammatory stimuli can regulate phospholipid biosynthesis in macrophages. Specifically, choline uptake and metabolism reduced inflammatory responses in response to LPS. The authors chose to test the role of choline metabolism on macrophages polarized during type 2 inflammatory settings.

In this manuscript, Ghorbani et al. demontrate that IL-4 can impact the phospholipid profile of macrophages upregulating choline uptake, metabolism and production of phosphatidylcholine. The authors go on to demonstrate that this pathway impacts Stat6 phosphorylation and promotes the upregulation of Retlna and Relma production in vitro and in vivo. Specifically, Relma production was attenuated after pharmacologic blockade of choline kinase in vitro and during helminth infection. However, RSM treatment also depleted macrophage populations in the peritoneum in uninfected and infected mice. This finding is not followed up on, and experimental caveats may be resulting in the phenotype. In contrast with diminished Relma production upon RSM treatment, macrophage expression of other M2 markers in vivo or in vitro is not significantly impacted, suggesting the choline pathway may only be regulating Relma, which is novel but underdeveloped. Overall the authors provide convincing data that choline metabolism can regulate Relma production, yet further experimentation is required to elucidate the specificity of choline activity on M2 gene expression as well address major concerns regarding the phenotype observed in the peritoneum.

**Part II – Major Issues: Key Experiments Required for Acceptance**

Reviewer #1: 1. It is not clear if choline metabolism is affecting the polarization process of M[IL-4] or cells that are already polarized and how specific this is to macrophage RELMa expression. It is important to clarify this aspect to support the manuscript’s conclusions.

a. In Fig. 3A-E and Fig. 4A-D, macrophages are pre-treated with HC3 or RMS to show effects of perturbing choline metabolism on polarization of M[IL-4]. In Fig. 3F, macrophages in the M[IL-4] state are treated to show effects on M[IL-4] phenotype. The authors should perform the assays in Fig. 3A-D and Fig. 4 in macrophages that are already M[IL-4] and should investigate IL-4Ra and RELMa protein expression.

b. The authors don’t show macrophage RELMa staining after H. polygyrus challenge (SFig. 5B-G) or in the peritoneal cavity (Fig. 7), making it hard to assess whether RSM treatment was impacting total macrophage populations or M[IL-4] macrophages, or just RELMa production in macrophages. The authors could shed light on this issue by transferring monocytes from RSM- or vehicle- treated mice into IL-4-complex-treated animals and investigating M[IL-4] development and RELMa production from transferred cells.

c. Krljanac et al. suggest that macrophage-derived RELMa supports host resistance to secondary N. brasiliensis infection (there are some caveats with the approaches used in these experiments, but the use of CD115iDTR with the RetnlaCre mice gets reasonably close to selectively deleting RELMa in macrophages). The authors should investigate whether RSM treatment and associated decreases in RELMa lead to impaired protection in N. brasiliensis secondary challenge.

2. There is no model to explain how inhibiting choline metabolism impacts M[IL-4] polarization. The authors show decreased pSTAT6 in macrophages pre-treated with choline metabolism inhibitors and then treated with IL-4 (Fig. 3E), but is there lower IL-4Ra expression on macrophages after choline metabolism inhibition? On page 8, the authors refer to data in Fig. 6 and SFig. 5 that show that the effects of choline metabolism on Retnla expression are IL-4-signaling-independent, but I can’t find these data (apologies if I missed them!). The authors don’t have to identify this mechanism – this seems beyond the scope of the current work to me - but as is, this issue is very murky in the text and should be discussed more clearly.

Reviewer #2: Main comments:

- The kinetics of choline metabolism inhibition in response to pharmacological drugs is quite puzzling. Why do you need 24h pre-incubation for seen an effect with a kinase inhibitor ? This part would have gained to be strengthen by using some genetic models, e.g. siRNA-mediated KD or use of BMDMs from whole-body KO mice (if available).

- The authors have used a knowledge-based selection of few alternatively-activated/M(IL4)/M2-like markers (notably Retlna/RELMalpha) to phenotype their macrophages in response to IL4 but would have gained to run at least few bulk RNAseq analyses in order to get an unbiased overview on the impact of choline metabolism inhibitors on macrophage identity.

- The alterations of helminth-induced peritoneal immune cell phenotypes observed in response to pre-treatment with choline metabolism inhibitor are quite difficult to understand, mechanistically-speaking, and could for sure not be ascertained to a specific impact of the drug on macrophages, as it may be suggested in between lines. Altogether, although interesting/puzzling, this part is also a bit confusing and shifts the story flow beyond macrophage choline metabolism.

Reviewer #3: 1. Figure 1B: Raw data of the western blots should be shown.

2. Figure 2: The authors indicate that the differences shown in panel B are significant. The authors should present the data to specifically indicate which lipids are significantly different in the bar graph.

3. Figure 3: Are there any differences in macrophage viability with HC3 or RSM treatment before or after IL-4 stimulation? The plots in Figure 3E give this impression. This panel is also lacking labels for each plot. Despite differences in STAT6 activation, Retlna is the only M2 marker regulated by choline metabolism, suggesting that this pathway may not act, at least exclusively, on IL-4 dependent pathways to regulate Retlna. The authors should provide data to develop the underlying mechanism by which choline may regulate Relma but not other established STAT6-dependent genes.

4. Figure 4: The authors suggest that the wound-healing effects are mediated by Relma, but provide no data to show actually demonstrating that Relma is active in this setting. While the use of Relma-/- macs would be ideal, the authors should at least compare these results to sups from non-polarized macs in the presence or absence of recombinant Relma treatment? Also the authors should cite the primary paper demonstrating the impact of Relma on wound healing (PMID: 26474656), not the review.

5. Figure 5: This figure is unclear. Did these animals receive IL-4 complexes to induce Relma production? If not, why would the peritoneal macs be expressing so much Relma at baseline? This data is not consistent with previous papers showing minimal Relma production by macrophages at steady-state (see figure 2A in PMID: 24101381 and figure 1A in PMID: 21566158). Can the authors reconcile these published results from their data? In panels C, D and E, what UMAP clusters are you referring to? Macrophages are well-known to undergo a response known as the Macrophage Disappearance Reaction (MDR) to certain stimuli under settings of inflammation (PMID: 7884305; PMID: 31048328). This reaction also coincides with the recruitment of monocytes. Therefore, the phenotype observed in the peritoneum after i.p. injecting mice with RSM may be secondary to inflammatory effects mediated by drug injection. This issue needs clarification with additional controls and raw flow cytometry data showing Relma production by peritoneal macs should be provided to substantiate the graphs.

6. Figure 6: The regulation of Relma by choline is interesting, independent of its effects on resistance to helminth infection. Given the lack of impact that RSM treatment has on worm burden and the unconvincing microscopy results (panel F), I suggest strengthening the peritoneal dataset including, as mentioned above, FACS data on Relma production following H. polygyrus infection as previously shown (PMID: 28334040).

7. Figure 7: The same issues are relevant here as in Figure 5, in terms of macrophage disappearance, but more striking. In addition to the concern that the changes in the peritoneum are simply due to injecting an inflammatory agent independent of its biological activity, increased clarity as to whether the authors are examining M2 markers on tissue-resident macrophages, monocyte-derived macrophages or total macrophages needs to be provided.

**Part III – Minor Issues: Editorial and Data Presentation Modifications**

Reviewer #1: 1. To better highlight key differences in the lipidomes of different macrophages, some panels from SFig. 2 could be moved to main Fig. 2.

2. Fig. 3E is missing the Vehicle, HC3, and RSM labels at the top of the flow plots.

3. In SFig. 3E, on my screen, there’s no convincing pAKT S473 signal. This could be a resolution issue. If not, then these data could be removed without altering any conclusions.

4. The scratch wound data in Fig. 4D are out of place with the focus on RELMa. There is no data to show that effects of choline metabolism inhibition on the ability of macrophage-conditioned media to promote wound healing is RELMa-dependent (unless this is known from previous literature). If this effect is RELMa-dependent, this should be stated. If not RELMa-dependent, these findings are still interesting, they should just be presented in a different context.

5. The authors should explain the significance of increased CD86 expression in SFig. 5.

6. The supplemental figure legends should state how many experiments were performed.

7. On page 6, as the authors discuss experiments with primary infection with H. polygyrus, they state that, “IL-4-polarized macrophages have been shown to play an important role in reducing helminth parasite burden; [29, 30]”. These references address the role of macrophages in N. brasiliensis secondary not H. polygyrus primary infection and they don’t address the role of RELMa. The authors later cite Anthony et al., which does establish a role for macrophages in protective immunity to H. polygyrus. In addition, it’s been shown that macrophage RELMa is important in reducing parasite burden in secondary N. brasiliensis infection (Krljanac et al.). The authors should clarify their discussion of the literature to make clear: 1) what was known previously, and 2) how their data fit with the literature.

8. The data shown in SFig. 5B-G do look as though there is some effect of RSM on protection, with trends to increased egg burdens, a significant difference at day 8, and a trend to increased adult worms. Are these observations consistent across replicate experiments?

9. The gut N. brasiliensis burdens in SFig. 7D are quite low for day 6. Can the authors comment?

10. In SFig. 7, the authors show a decrease in RELMa staining from lung epithelial cells in mice treated with RSM and infected with N. brasiliensis. What about macrophages?

Reviewer #2: Minor comments:

- Couple of typos should be corrected throughout the manuscript, including some in the abstract (“by injection IL-4”, “no difference intestinal worm”)

- Some rephrasing are also required, e.g. “using similar but still differential study designs”

- Please clarify the rationale for measuring Ser473-pAkt

- The authors claim that “in vivo inhibition of choline metabolism restructures endogenous tissue-resident macrophage phenotype”. However, it is unclear how they define “tissue-residency”. MHC-II and/or TIM-4 expression coupled to F4/80 is generally used to discriminate newly recruited monocyte-derived macrophages versus long-lasting tissue-resident macrophage populations.

- Statistical analyses on Figure 3F are missing

Reviewer #3: (No Response)

PLOS authors have the option to publish the peer review history of their article (what does this mean?). If published, this will include your full peer review and any attached files.

Reviewer #1: No

Reviewer #2: No

Reviewer #3: No
---

## [Decision Letter · Decision Letter 1]

7 Aug 2023

Dear Dr Fullerton,

Thank you very much for submitting your manuscript "Choline metabolism underpins macrophage IL-4 polarization and RELMα up-regulation in helminth infection" for consideration at PLOS Pathogens. As with all papers reviewed by the journal, your manuscript was reviewed by members of the editorial board and by several independent reviewers. The reviewers appreciated the attention to an important topic. Based on the reviews, we are likely to accept this manuscript for publication, providing that you modify the manuscript according to the review recommendations.

One reviewer has raised an important issue that was not addressed from the first review, and that is the number of experimental replicates in the data in the supplemental experiments (See reviewer 1). I am requesting that this experiment be removed from the manuscript if a replicate experiment can not be added that has the same trends and significance. I agree with the Reviewer that the standard in the field is a minimum of duplication, but ideally 3 biologically independent experiments. THere is also a minor point in figure 4 that needs to be fixed for clarity of readers listed by Reviewer 2.

Sincerely,

Keke C Fairfax, PhD

Guest Editor

PLOS Pathogens

P'ng Loke

Section Editor

PLOS Pathogens

Kasturi Haldar

Editor-in-Chief

PLOS Pathogens

orcid.org/0000-0001-5065-158X

Michael Malim

Editor-in-Chief

PLOS Pathogens

orcid.org/0000-0002-7699-2064

One reviewer has raised an important issue that was not addressed from the first review, and that is the number of experimental replicates in the data in the supplemental experiments (See reviewer 1). I am requesting that this experiment be removed from the manuscript if a replicate experiment can not be added that has the same trends and significance. I agree with the Reviewer that the standard in the field is a minimum of duplication, but ideally 3 biologically independent experiments. THere is also a minor point in figure 4 that needs to be fixed for clarity of readers listed by Reviewer 2.

Reviewer Comments (if any, and for reference):

Reviewer's Responses to Questions

**Part I - Summary**

Reviewer #1: The authors have answered the majority of my queries and present a very nice study. One outstanding issue remains as regards inclusion of experiments that were only performed a single time (see below).

Reviewer #2: The authors have satisfactorily addressed most of the point form this reviewer, notably by the inclusion of bulk RNAseq data, and significantly improved the quality of their manuscript. I have no further major points of concern.

Reviewer #3: The authors have made considerable effort to improve the manuscript and have sufficiently addressed my concerns.

**Part II – Major Issues: Key Experiments Required for Acceptance**

Reviewer #1: My original Minor Point #7 requested that the number of replicate experiments performed for all supplemental figures be added to the legends. In the authors' response, they indicated that these experiments were only performed once, as supportive of the main findings and not, "stand alone."

I appreciate that these are lengthy experiments and difficult to do multiple times. However, presenting data from experiments only performed once is not in line with standards in the field. Independent, replicate experiments should be performed at the very least twice, and ideally, 3 times. If these experiments cannot be repeated, the data should not be presented, in my opinion. Even if the data are confirmatory of other findings and in the supplement, I feel that the standards for rigor should be maintained.

I will defer final judgment on that to the Editorial team. However, if the Editorial team feels that inclusion of data from single experiments is acceptable, I request that this fact be very clearly acknowledged in the legends, and in the text itself as well.

Reviewer #2: (No Response)

Reviewer #3: (No Response)

**Part III – Minor Issues: Editorial and Data Presentation Modifications**

Reviewer #1: (No Response)

Reviewer #2: The y-axis scale of the Seahorse data should be similar for allowing easy comparisons (Fig 4C&E)

Reviewer #3: (No Response)

PLOS authors have the option to publish the peer review history of their article (what does this mean?). If published, this will include your full peer review and any attached files.

Reviewer #1: No

Reviewer #2: **Yes: **Bruno Guigas

Reviewer #3: No

Figure Files:

Data Requirements:

Reproducibility:

References:

---

## [Editor Report · Decision Letter 2]

5 Sep 2023

Dear Dr Fullerton,

We are pleased to inform you that your manuscript 'Choline metabolism underpins macrophage IL-4 polarization and RELMα up-regulation in helminth infection' has been provisionally accepted for publication in PLOS Pathogens.

Best regards,

Keke C Fairfax, PhD

Guest Editor

PLOS Pathogens

P'ng Loke

Section Editor

PLOS Pathogens

Kasturi Haldar

Editor-in-Chief

PLOS Pathogens

orcid.org/0000-0001-5065-158X

Michael Malim

Editor-in-Chief

PLOS Pathogens

orcid.org/0000-0002-7699-2064
---

## [Editor Report · Acceptance letter]

18 Sep 2023

Dear Dr Fullerton,

We are delighted to inform you that your manuscript, "Choline metabolism underpins macrophage IL-4 polarization and RELMα up-regulation in helminth infection," has been formally accepted for publication in PLOS Pathogens.

Best regards,

Kasturi Haldar

Editor-in-Chief

PLOS Pathogens

orcid.org/0000-0001-5065-158X

Michael Malim

Editor-in-Chief

PLOS Pathogens

orcid.org/0000-0002-7699-2064